# Infinite-Dimensional Generative Diffusions via Doob's h-Transform

Thorben Pieper-Sethmacher [* 1]   Daniel Paulin [* 1]

## Abstract

This paper introduces a rigorous framework for defining generative diffusion models in infinite dimensions via Doob's h-transform. Rather than relying on time reversal of a noising process, a reference diffusion is forced towards the target distribution by an exponential change of measure. Compared to existing methodology, this approach readily generalises to the infinite-dimensional setting, hence offering greater flexibility in the diffusion model. The construction is derived rigorously under verifiable conditions, and bounds with respect to the target measure are established. We show that the forced process under the changed measure can be approximated by minimising a score-matching objective and validate our method on both synthetic and real data.

## 1. Introduction

Over the past decade, generative diffusion models (Sohl-Dickstein et al., 2015; Ho et al., 2020; Song et al., 2021) have established themselves as a state-of-the-art paradigm in generative models, finding widespread applications in sampling data such as text (Li et al., 2022), image (Dhariwal & Nichol, 2021; Rombach et al., 2022) and video (Ho et al., 2022; Singer et al., 2023).

In many such applications, the data is treated as a finite-dimensional discretisation of inherently infinite-dimensional objects. Typical examples include images and geometric shapes (Dupont et al., 2021), spatiotemporal physical processes (Shu et al., 2023; Rühling Cachay et al., 2023) and function-space measures arising in Bayesian inverse problems (Stuart, 2010; Oliviero-Durmus et al., 2025).

In recent work, there has been an effort to adapt generative diffusion models to such an infinite-dimensional setting

[1]College of Computing and Data Science, Nanyang Technological University, Singapore. Correspondence to: Thorben Pieper-Sethmacher <thorben.ps@ntu.edu.sg>, Daniel Paulin <Daniel.paulin@ntu.edu.sg>.

*Proceedings of the 43rd International Conference on Machine Learning*, Seoul, South Korea. PMLR 306, 2026. Copyright 2026 by the author(s).

(Kerrigan et al., 2023; Franzese et al., 2023a; Lim et al., 2023; Pidstrigach et al., 2024; Lim et al., 2025; Hagemann et al., 2025). See also (Franzese & Michiardi, 2025) for a survey on the literature in this direction.

In this infinite-dimensional regime, discretising the target measure and subsequently applying methods developed for finite-dimensional data might degenerate and fail to converge to the correct target distribution as the dimension of the discretisation (or the *data resolution*) tends to infinity.

In contrast, designing generative diffusion models directly in the infinite-dimensional setting and only discretising during the implementation step leads to methodology that remains robust to the state dimension of the target measure.

This *discretise last* paradigm has been prevalent in the literature for Bayesian inverse problems for over a decade, leading to well-posed formulations on function spaces and discretisation-independent algorithms (Stuart, 2010; Beskos et al., 2011; Cotter et al., 2013; Hairer et al., 2014).

Current generalisations of infinite-dimensional diffusion models primarily focus on the noising-denoising paradigm. This is typically implemented via one of two main approaches: discrete-time sequences that vary noise scales across data samples (Kerrigan et al., 2023; Lim et al., 2025), or continuous-time formulations that reverse the dynamics of a stochastic differential equation (SDE) (Franzese et al., 2023a; Pidstrigach et al., 2024; Hagemann et al., 2025; Lim et al., 2023).

Recall that, in this noising-denoising framework (Song et al., 2021), sampling from the target measure relies on reversing a forward noising SDE, whose marginals are designed to converge to a tractable (typically Gaussian) reference measure. Consequently, time-reversal based diffusion models exhibit fundamental pathologies when the noising process fails to converge in time to the prescribed reference measure (De Bortoli et al., 2021; Franzese et al., 2023b).

This work introduces a class of infinite-dimensional generative diffusion models by *forcing* a reference process $X$, obtained as the mild solution to an infinite-dimensional SDE, to the target measure $\mu$ at some final time $T > 0$ by means of a change of measure known as *Doob's h-transform*. Crucially, this construction is valid for any $T$, thereby circumventing the need for long-time convergence of a noising

process. Under suitable assumptions, the forced process $X^h$ under the changed measure satisfies yet another SDE with an additive steering term dependent on the $h$-function of choice, which can be approximated by minimising a score-matching-type objective.

For finite-dimensional data, related approaches have been studied in (De Bortoli et al., 2021; Peluchetti, 2023b;a; Didi et al., 2023; Wu et al., 2023; Nguyen et al., 2025; Chang et al., 2026). In the infinite-dimensional setting, (Park et al., 2024) have recently considered $h$-transformed diffusion models from an optimal control perspective. In this approach, the focus lies on learning bridges between prescribed endpoint distributions, whereas we consider the task of learning a generative diffusion process. In addition, our results are more general in that they allow for time-dependent and nonlinear drifts. Moreover, we provide a rigorous derivation by stating explicit conditions under which the $h$-transformed process is well defined and approximable, together with bounds between the sampling and target measures.

### 1.1. Challenges and Approach

Recall that, in the noising-denoising SDE framework, samples of the target $\mu$ are progressively noised by a process

$$\mathrm{d}X_t = f(t, X_t)\,\mathrm{d}t + b(t)\,\mathrm{d}W_t, \quad t \in [0, T],$$

with $X_0 \sim \mu$. Under suitable assumptions, $X_t$ converges to a tractable, invariant reference measure $\nu$. Assuming that $X_T \sim \nu$ in law, the data distribution is then sampled from by solving the reverse-time SDE

$$\mathrm{d}\bar{X}_t = \left[-f(t, \bar{X}_t) + b^2(t)s(T - t, \bar{X}_t)\right]\mathrm{d}t + b(t)\,\mathrm{d}W_t,$$

with $\bar{X}_0 \sim \nu$ and the score $s(t, x) = \mathrm{D}_x \log p_t(x)$ approximated by minimising a denoising score matching objective (Hyvärinen, 2005; Vincent, 2011).

One faces a number of challenges when lifting this to an infinite dimensional Hilbert space $H$. Firstly, the lack of Lebesgue measure on $H$ raises the question on how the marginal $p_t(x)$ is to be understood. Typically, one either constructs a suitable Gaussian reference measure as in (Lim et al., 2025) or, if the drift is linear, bounded and constant, abstracts the score as a conditional expectation (Pidstrigach et al., 2024; Baldassari et al., 2023).

Secondly, naively taking $W$ to be a cylindrical Wiener process on $H$ - the limiting object of finite-dimensional Wiener process - renders most noising processes ill-defined. Instead, $W$ needs to inject noise that is *preconditioned* to match the regularity of the target measure $\mu$ by some covariance operator $C$. One typically distinguishes two fundamentally different regimes; given an associated Gaussian measure $\nu = \mathcal{N}(0, C)$ on $H$, one assumes that either (A) $\mu \ll \nu$

or (B) $\mu$ to be supported on the Cameron-Martin space $H_\nu = C^{\frac{1}{2}}(H)$ of $\nu$. Notice that, since the Cameron-Martin space $H_\nu$ is a $\nu$-null set in the infinite-dimensional setting, these regimes are mutually exclusive for any given $\nu$. Heuristically speaking, situation (A) typically corresponds to target measures $\mu$ whose sample paths are as *rough* as the reference Gaussian measure. These appear, for example, in Bayesian inverse problems (Stuart, 2010; Baldassari et al., 2023; Oliviero-Durmus et al., 2025) and data assimilation (Bain & Crisan, 2009). To the best of our knowledge, in the context of generative diffusions, this regime has so far only been considered in (Pidstrigach et al., 2024) and (Baldassari et al., 2023).

On the other hand, situation (B) is applicable in settings in which $\mu$ lies on a manifold, possibly lower-dimensional, of *smooth* samples, as is common in applications involving image and video data (Pope et al., 2021; Thornton et al., 2022). Generative diffusions under this assumption have been treated in (Pidstrigach et al., 2024; Lim et al., 2025; Franzese et al., 2023a; Hagemann et al., 2025; Lim et al., 2023).

Lastly, proving the existence of the time reversed process is notoriously difficult. One either needs to assume simple, bounded dynamics or assume a number of hard-to-verify assumptions, see (Franzese & Michiardi, 2025) for a detailed discussion on this issue.

In our approach, we circumvent the need of a time reversal by instead *forcing* a reference process

$$\mathrm{d}X_t = f(t, X_t)\,\mathrm{d}t + b(t)\,\mathrm{d}W_t, \quad t \in [0, T],$$

initiated at some tractable measure $\mu_0$, to hit the target measure $\mu$ at time $T$. Forcing is carried out by a change of measure known as *Doob's h-transform*, a well-known technique to condition SDEs for example in the context of diffusion bridges (Baudoin, 2002; Schauer et al., 2017; Heng et al., 2025). In the infinite-dimensional context, it has been shown to provide well-defined conditioning for a wide class of processes (Fuhrman, 2003; Baker et al., 2024; Pieper-Sethmacher et al., 2025a). The forced process satisfies the SDE

$$\mathrm{d}X_t^h = [f(t, X_t^h) + b^2(t)s(t, X_t^h)]\,\mathrm{d}t + b(t)\,\mathrm{d}W_t,$$

where $s(t, x) = \mathrm{D}_x \log h(t, x)$ with $h$ defining the change of measure. By leveraging the fact that the $h$-transform is applicable for unbounded operators $f$, forced generative diffusions can be constructed whose transition kernels are absolutely continuous with respect to a Gaussian reference measure if $\mu$ satisfies either of the data regimes (A) or (B) above.

The dynamics of $X^h$ closely mirror the dynamics of the time reversed process, though a fundamental difference is

that $X_T^h$ is ensured to reach the target measure $\mu$, regardless of the time horizon $T$. This is partly due to effect on the measure transformation on the initial distribution $\mathrm{d}\mu_0^h = h(0, x)\,\mathrm{d}\mu_0$, which corrects for a mismatch between the prior $\mu_0$ and the (backwards) noised data distribution if $T$ is not sufficiently large. As is typical, the functions $h(t, x)$ and $s(t, x)$ are generally unknown. However, given samples from the target $\mu$, $s(t, x)$ can be approximated by minimising a score-matching-type objective. After training, new samples of $\mu$ can then be generated by (i) Langevin sampling from $\mu_0^h$ based on the score $h(0, x)$, and (ii) subsequently forward simulating the forced process $X^h$.

## 1.2. Contribution

We introduce a principled framework for defining generative diffusion models in infinite dimensions based on Doob's $h$-transform. This circumvents the need for time reversal and instead forces a reference diffusion to the target measure. Unlike existing noising-denoising approaches, our method is well-defined for arbitrary time horizons and avoids pathologies caused by the lack of convergence of a noising process. A rigorous derivation is provided, establishing the existence of the $h$-transformed process for a wide class of infinite-dimensional reference diffusions under verifiable conditions. Moreover, we show that the $h$-transformed process can be learned via a score-matching objective and derive bounds between the induced sampling measure and the target. The effectiveness of our approach is demonstrated on both synthetic and real-world infinite-dimensional data.

## 2. Setup

Let $H$ be an infinite-dimensional Hilbert space, equipped with its Borel algebra $\mathcal{B}(H)$ and let $\mu$ be a measure on $H$. Given a training data set $y_1, \dots y_N \overset{\text{i.i.d.}}{\sim} \mu$, we consider the task of learning to sample from the target $\mu$. To derive our theory, we assume $\mu$ to satisfy the following condition, which corresponds to situation (A) in Section 1.1.

**Assumption 2.1.** There exists a Gaussian measure $\nu = \mathcal{N}(0, C)$ on $H$ such that $\nu$ dominates $\mu$.

We define a generative diffusion by *forcing* or (*steering*) a reference diffusion process $X$ to equal, in law, $\mu$ at an arbitrary time $T > 0$. For this, introduce the infinite-dimensional SDE

$$\mathrm{d}X_t = [A(t)X_t + F(t, X_t)]\,\mathrm{d}t + B(t)\,\mathrm{d}W_t, \quad (1)$$

where $A = (A(t))_{t \geq 0}$ is a family of densely defined operators with a common domain $D_A$, generating an evolution family $U = (U(t, s))_{0 \leq s \leq t \leq T}$. Moreover, $F(t, x)$ is a Lipschitz continuous non-linearity (both in $t$ and $x$), $B$ a family of bounded, linear operators with dense range in $H$ and $W$ a cylindrical Wiener process on $H$, defined on a stochastic ba-

sis $(\Omega, \mathcal{F}, (\mathcal{F}_t)_t, \mathbb{P})$. It is assumed throughout that, for any initial value $x_0$, Equation (1) admits a unique mild solution $(X_t)_{t \in [0, T]}$ satisfying

$$X_t = U(t, 0)x_0 + \int_0^t U(t, s)F(s, X_s)\,\mathrm{d}s + \int_0^t U(t, s)B(s)\,\mathrm{d}W_s. \quad (2)$$

For details on the existence and uniqueness of $X$ in the general case, see (Da Prato & Zabczyk, 2014; Veraar, 2010; Knäble, 2011; Cerrai & Lunardi, 2025). Existence and uniqueness for relevant diffusions in the context of generative diffusions is shown in Section 5.

## 3. Forcing via Doob's $h$-transform

This section introduces the appropriate measure transformation known as *Doob's h-transform* under which the process $X$ is forced towards the distribution $\mu$. Throughout, let $X$ be initiated with $X_0 \sim \mu_0$ for some Borel measure $\mu_0$. The following assumption will be needed.

**Assumption 3.1.** The mild solutions $X$ to (1) admit transition densities $p(s, x; t, y)$ with respect to $\nu$, defined by

$$\mathbb{P}(X_t \in A \mid X_s = x) = \int_A p(s, x; t, y)\,\mathrm{d}\nu(y),$$

for all $x \in H, A \in \mathcal{B}(H)$ and $0 \leq s < t \leq T$. Moreover, the transition- and marginal densities $p_t(x)$ are such that

$$h(t, x) = \int_H \frac{p(t, x; T, y)}{p_T(y)}\,\mathrm{d}\mu(y), \quad t < T, \quad (3)$$

is well defined with Fréchet derivative $\mathrm{D}_x h(t, x)$ for any $t < T$ of at most polynomial growth in $x \in H$.

*Remark* 3.2. Assumption 3.1 might at first seem restrictive, as absolute continuity of measures in the infinite-dimensional setting is typically hard to obtain. However, as will be shown in Section 5, a wide class of diffusions that satisfy Assumption 3.1 can be constructed by leveraging the fact that forced processes can be derived for SDEs with unbounded drift operators.

To force the diffusion $X$ towards the target measure $\mu$ at time $T$, we apply a version of Doob's $h$-transform with the choice of $h$ as defined in (3). The following theorem makes this precise.

**Theorem 3.3.** *There exists a unique measure $\mathbb{P}^h$ on $\mathcal{F}_T$, defined by*

$$\mathrm{d}\mathbb{P}_t^h = h(t, X_T)\,\mathrm{d}\mathbb{P}_t, \quad t < T,$$

*such that $\mathbb{P}^h(X_T \in A) = \mu(A)$ for all $A \in \mathcal{B}(H)$. More-*

*over, $X$ under $\mathbb{P}^h$ is a mild solution to*

$$
\begin{aligned}
\mathrm{d}X_t^h &= \left[A(t)X_t^h + F(t, X_t^h) + (BB^*)(t)s(t, X_t^h)\right] \mathrm{d}t \\
&\quad + B(t)\,\mathrm{d}W_t^h, \\
X_0^h &\sim \mu_0^h(x),
\end{aligned}
\tag{4}
$$

*where $s(t, x) = \mathrm{D}_x \log h(t, x)$, $\mathrm{d}\mu_0^h(x) = h(0, x)\,\mathrm{d}\mu_0(x)$ and $W^h$ is a $\mathbb{P}^h$-cylindrical Wiener process.*

Following Theorem 3.3, if $s(t, x)$ is known, new samples of $\mu$ can be generated by (i) sampling from $x_0 \sim \mu_0^h$, for example through an infinite-dimensional Langevin sampler (Cotter et al., 2013), and (ii) subsequently evolving $x_0$ according to the forced SDE (4).

*Remark* 3.4. The initial step (i) is a fundamental feature of our proposed methodology and prevents pathologies prevalent in generative diffusion models due to insufficient noising at time $T$. Notice that sampling of $X^h$ can be characterised in two equivalent ways. In addition to solving Equation (4), samples of $X$ under $\mathbb{P}^h$ can be obtained by the following disintegration (cf. Lemma B.3): (a) draw $y \sim \mu$ from the target, and then (b) sample the *infinite-dimensional diffusion bridge* $X_t^y := X_t \mid X_T = y$ under $\mathbb{P}$.

The diffusion bridge $X_t^y$ can be understood as *noising* the samples $y$ of the target $\mu$ *backwards in time* to $X_0^y$. This implies that, if the reference diffusion $X$ is mixing and $T$ is sufficiently large, $p(0, x; T, y)$ may be assumed to be approximately independent of $x$ and $h(0, \cdot) \equiv 1$. In that case, our methodology closely resembles the typical noising-denoising framework.

However, if this assumption is violated, there is a mismatch between the noised data distribution and the assumed prior $\mu_0$. In that case, sampling from $\mu_0^h$ through a Langevin sampler with score $s(0, x) = \mathrm{D}_x \log h(0, x)$ corrects this mismatch. This ensures that step (ii) draws exact samples of $\mu$ at time $T$, *independently of the choice of $T$*. This contrasts with time-reversed generative diffusions, in which insufficient noising time $T$ causes biased and, in extreme cases, degenerate sampling.

## 4. Approximation of Doob's h-transform

The previous section introduced the appropriate choice of the $h$-function to steer $X$ towards $\mu$ based on Doob's h-transform. As is typical, except for a limited number of special cases, $h(t, x)$ and hence the steering function $s(t, x)$ appearing in the transformed diffusion (4) are unknown. However, given samples $y_1, \ldots y_N \overset{\mathrm{i.i.d.}}{\sim} \mu$, the steering function can be approximated by minimising a loss that resembles the score-matching objective of the noising-denoising approach (Hyvärinen, 2005; Vincent, 2011).

For this, let $s_\theta$ be some parametrised approximation of $s$ for

some $\theta$ in a parameter space $\Theta$ and let $X^\theta$ be a diffusion process satisfying

$$
\begin{aligned}
\mathrm{d}X_t^\theta &= [A(t)X_t^\theta + F(t, X_t^\theta)]\,\mathrm{d}t \\
&\quad + (BB^*)(t)s_\theta(t, X_t^\theta)\,\mathrm{d}t \\
&\quad + B(t)\,\mathrm{d}W_t,
\end{aligned}
\tag{5}
$$

with initial distribution $X_0^\theta \sim \mu_0$.

Denote by $\mathbb{X}^h$ and $\mathbb{X}^\theta$ the path measures of $X^h$ and $X^\theta$ on $C(0, T; H)$. The following proposition defines a loss $\mathcal{L}(\theta)$ whose minimisation is equivalent to minimising the Kullback-Leibler divergence from $\mathbb{X}^h$ to $\mathbb{X}^\theta$.

**Proposition 4.1.** *Define the loss function*

$$
\mathcal{L}(\theta) = \int_0^T \mathbb{E}^h |s_\theta(t, X_t) - \mathrm{D}_{x_t} \log p(t, X_t; T, X_T)|_{B_t^*}^2 \,\mathrm{d}t,
\tag{6}
$$

*where $|x|_{B_t^*} = |B^*(t)x|$. Then $\theta^\star = \arg\min_\theta \mathcal{L}(\theta)$ if and only if $\theta^\star = \arg\min_\theta D_{\mathrm{KL}}(\mathbb{X}^h \| \mathbb{X}^\theta)$.*

*Remark* 4.2. In practice, the loss function $\mathcal{L}(\theta)$ is to be approximated by a Monte Carlo scheme based on samples of the path $X$ under $\mathbb{P}^h$. As noted in Remark 3.4, these can be drawn by simulating diffusion bridges $X^{y_i}$ for the given data samples $y_1, \ldots y_N$. If the reference diffusion (1) is linear and the initial distribution $\mu_0$ is either a Dirac measure or a Gaussian measure, absolutely continuous with respect to $\nu$, then $X^y$ is the tractable infinite-dimensional Ornstein-Uhlenbeck bridge, which can be sampled directly (Goldys & Maslowski, 2008). Moreover, since the transition density of $X$ is Gaussian, the score $\mathrm{D}_x \log p(t, x; T, y)$ can be computed in closed-form.

In the case that (1) is non-linear, the conditioned process $X^y$ is itself intractable. In that case, weighted samples of the non-linear diffusion bridge $X^y$ can be drawn based on *guided proposals* (Schauer et al., 2017; Pieper-Sethmacher et al., 2025b). Moreover, $\mathrm{D}_x \log p(t, x; T, y)$ may be approximated numerically, for example based on an EM-scheme as done in (Heng et al., 2025). The loss $\mathcal{L}(\theta)$ can then be approximated via importance sampling.

In order to fully approximate $\mathbb{X}^h$, we require the class of parametrised approximations $\{s_\theta : \theta \in \Theta\}$ to be sufficiently large as stated in the following.

**Assumption 4.3.** The function $s$ is continuous and there exists some $\theta \in \Theta$ such that $s_\theta(t, x) = s(t, x)$ for every $t \in [0, T]$, $\nu$-a.e. $x \in H$.

**Corollary 4.4.** *Let Assumption 4.3 hold and let $X^\theta$ satisfy (5) such that the law of $X_0^\theta$ has score $s_\theta(0, x)$ with respect to $\mu_0$. Then $\theta^\star = \arg\min_\theta \mathcal{L}(\theta)$ if and only if $\mathbb{X}^{\theta^*} = \mathbb{X}^h$.*

## 5. VP-SPDE

Inspired by the success of the *Variance Preserving (VP)* SDE proposed in (Song et al., 2021), we introduce in this section the *VP-SPDE*. The VP-SPDE mimics the temporal noise-scheduling of the VP-SDE, while accounting for the spatial structure of the target measure, as is necessitated by the infinite-dimensional setting.

For this, let $\beta(t)$ be some bounded, positive function and define the operators $A = \frac{1}{2}C^{-\gamma}$ and $Q = C^{1-\gamma}$ for some $\gamma \in (0,1]$. Denote by $X$ the mild solution to the inhomogeneous Ornstein-Uhlenbeck equation

$$
\begin{aligned}
\mathrm{d}X_t &= -\beta(t)AX_t\,\mathrm{d}t + \sqrt{\beta(t)Q}\,\mathrm{d}W_t, \\
X_0 &\sim \mathcal{N}(0, C).
\end{aligned}
\tag{7}
$$

Notice that the covariance operator $C$, predetermined by Assumption 2.1, is a trace-class operator. This renders $C^{-\gamma}$ an unbounded, densely defined operator on $H$, hence motivating the terminology of $X$ as the VP-SPDE.

**Proposition 5.1.** *There exists a unique mild solution $X$ to the VP-SPDE* (7). *Moreover, $X$ satisfies Assumption 3.1.*

The immediate corollary of this result and Theorem 3.3 is that the forced process via Doob's $h$-transform exists for (7).

**Corollary 5.2.** *Let $X$ be the mild solution to* (7)*, then $h(t,x)$ defined as in* (3) *exists with a well-defined Fréchet derivative $D_x h(t,x)$ for any $t < T$ of at most polynomial growth in $x \in H$. In particular, there exists a unique measure $\mathbb{P}^h$ on $\mathcal{F}_T$, defined by*

$$
\mathrm{d}\mathbb{P}^h_t = h(t, X_T)\,\mathrm{d}\mathbb{P}_t, \quad t < T,
$$

*such that $\mathbb{P}^h(X_T \in A) = \mu(A)$ for all $A \in \mathcal{B}(H)$. Moreover, $X^h$ under $\mathbb{P}^h$ is a mild solution to*

$$
\begin{aligned}
\mathrm{d}X^h_t &= \left[-\beta(t)AX^h_t + \beta(t)Qs(t, X^h_t)\right]\mathrm{d}t \\
&\quad + \sqrt{\beta(t)Q}\,\mathrm{d}W^h_t, \\
X^h_0 &\sim \mu^h_0(x),
\end{aligned}
\tag{8}
$$

*where $s(t,x) = D_x \log h(t,x)$, $\mathrm{d}\mu^h_0(x) = h(0,x)\,\mathrm{d}\mu_0(x)$, $\mu_0 = \mathcal{N}(0,C)$ and $W^h$ is a $\mathbb{P}^h$-cylindrical Wiener process.*

*Remark* 5.3. The parameter $\gamma$ controls the roughness of the noising in our model, with larger $\gamma$ corresponding to rougher noise. However, for the diffusion $X$ to remain well-defined, any such rough noise needs to be smoothened out by the drift operator $A = \frac{1}{2}C^{-\gamma}$. The white noise case $\gamma = 1$ was pointed out as a possible noising process candidate in (Pidstrigach et al., 2024). As noted there, this leads to a strong smoothing property of the semigroup generated by $A$, under which high-frequency information of the data is lost almost immediately. In our experiments, we have indeed found

that choices of $\gamma \approx 0$ tend to perform better, although further research on how the choice of $\gamma$ influences the models performance based on the structure of $\mu$ is needed.

*Remark* 5.4. The variance preserving property of the VP-SPDE in this context needs to be understood from the perspective of the time-reversed noising of the diffusion bridge $X^y_t$. Given that $X_0 \sim \mathcal{N}(0,C)$, it holds that $X^y_t$ is Gaussian with mean $U(T,t)y$ and covariance operator $C\left[I - U^2(T,t)\right]$, where $U$ is the diagonal evolution family with eigenvalues

$$
u(T,t)_j = \exp\left(-\frac{1}{2}c_j^{-\gamma}\int_t^T \beta(r)\,\mathrm{d}r\right).
$$

Hence, if the target $\mu$ has covariance operator $C$, integrating out $y \sim \mu$ in $X^y_t$ shows that the marginals $X^h_t$ of the forced process have constant covariance operator $C$.

As a consequence of Proposition 5.1 and the Girsanov theorem, we get the following additional result.

**Corollary 5.5.** *Let $F$ be a bounded and Fréchet differentiable function with bounded derivative and let $X$ be the mild solution to* (1) *with $A(t) = -\frac{1}{2}\beta(t)C^{-\gamma}$ and $B(t) = \sqrt{\beta(t)C^{(1-\gamma)}}$. Then $X$ satisfies Assumption 3.1.*

In principle, Corollary 5.5 allows us to construct generative diffusion models based on non-linear reference diffusions. However, due to the additional complexity of estimating $\mathcal{L}(\theta)$ and the widespread success of noising-denoising frameworks based on linear SDEs, we focus our attention in our applications on the linear VP-SPDE case. We give a detailed summary of the algorithmic methodologies to train and sample the forced generative VP-SPDE in the supplementary material, Section F.

## 6. Bounds to the target measure

Our approach consists of two parts:

1. Training the score estimator $s_\theta(t,x)$ based on the sample-based empirical estimator of the loss function 6, as explained in Remark 4.2.

2. Simulation of the forced process (8). This starts by approximate initialisation at $\mu^h_0$, for example through an infinite-dimensional Langevin sampler such as (Cotter et al., 2013), initiated for the VP-SPDE (7) at $\mu_0 = \mathcal{N}(0,C)$. After this, we use a semi-implicit Euler scheme for simulating (8).

The following result characterises the Wasserstein error of the samples produced by this algorithm, and disentangles the different sources of error (numerical integrator, error in score estimation, and error in initialisation via Langevin steps).

**Proposition 6.1.** *Consider the training procedure outlined above, based on* (8)*. Assume that $\beta(t)Qs(t,x) = \beta(t)Q\,\mathrm{D}_x \log h(t,x)$ is Lipschitz on $H$ with time uniform constant, i.e.,*

$$\|\beta(t)Q(s(t,x) - s(t,y))\|_H \le L\|x - y\|_H,$$

*and that the same condition also holds for the fitted score $s_\theta(t,y)$ with the same constant $L$.*

*Assume that the initial distribution used $\hat{\mu}_0^h$ (typically obtained via Langevin steps from $\mu_0$) approximates $\mu_0^h$ well in 2-Wasserstein distance, $\mathcal{W}_2(\hat{\mu}_0^h, \mu_0^h) \le \varepsilon_{\mathrm{Init}}$.*

*Suppose that the process* (8) *is discretized using the semi-implicit Euler method defined in (3.55) of (Kruse, 2014). Suppose that the assumptions of Theorem 3.14 of (Kruse, 2014) hold for the semi-implicit Euler discretization of* (8)*. Then*

$$\mathcal{W}_2(\mu_{\mathrm{data}}, \mu_{\mathrm{sample}}) \le \left( \varepsilon_{\mathrm{Init}} + \varepsilon_{\mathrm{Loss}}^{1/2} \right) \exp\left( LT \right) + \varepsilon_{\mathrm{Num}}. \tag{9}$$

*Here $\varepsilon_{\mathrm{Loss}}$ denotes the value of the loss function* (6) *at the trained score $s_\theta(x,t)$, and $\varepsilon_{\mathrm{Num}}$ is the error due to the numerical integration procedure ($\Delta t$ is time discretization, $h_{\mathrm{space}}$ is spatial discretization),*

$$\varepsilon_{\mathrm{Num}} = O(\Delta t)^{1/2} + O(h_{\mathrm{space}}).$$

*Remark* 6.2. In our setting, $h_{\mathrm{space}} \in (0,1]$ depends on the number of dimensions that we implement in the discretization of the infinite dimensional operator $A$, and it is defined precisely in Section 3.2 of (Kruse, 2014). Under the Assumption 3.3 of (Kruse, 2014), which is also assumed in our proposition, $h_{\mathrm{space}}$ tends to 0 as we increase the dimension of the discretization.

*Remark* 6.3. Compared with Theorem 14 of (Pidstrigach et al., 2024), our result does not include the $W_2(\mu_{data}, \mathcal{N}(0,C)) \exp(-T/2 + L^2 T/4)$ term, which grows with $T$ and may be significant when the data distribution is highly non-Gaussian. Instead of this, we have $\epsilon_{\mathrm{Init}} \exp(LT)$, which may be reduced by doing additional Langevin steps to better approximate $\mu_0^h$. In our numerical simulations, $\mu_0^h$ was close to $\mu_0$, and we could keep the number of Langevin steps small.

### 6.1. The case that $\mathrm{supp}(\mu) \subset H_\nu$

Our methodology is derived under the assumption that $\mu \ll \nu$ for some Gaussian reference measure $\nu = \mathcal{N}(0,C)$. In many applications, one instead targets a measure $\mu$ which models data that is inherently smoother than $\nu$ and is supported on the Cameron-Martin space $H_\nu = C^{\frac{1}{2}}(H)$ of $\nu$. This corresponds to situation (B) introduced in Section 1.1.

In that case, the proposed approach remains valid after an arbitrarily small noising of the data through the SDE

$$\mathrm{d}Y_t = -\tfrac{1}{2}C^{-\gamma}Y_t\,\mathrm{d}t + \sqrt{C^{1-\gamma}}\,\mathrm{d}W_t, \quad t \in [0, \varepsilon], \\ Y_0 \sim \mu, \tag{10}$$

and by replacing the original target measure $\mu$ with the slightly regularized measure $\mu_\varepsilon$, defined as the law of $Y_\varepsilon$. Heuristically speaking, this is equivalent to the common practice of running reverse-time denoising diffusions up to a time $\varepsilon > 0$.

As the following Lemma shows, this recovers Assumption 2.1 for any arbitrary $\varepsilon > 0$.

**Lemma 6.4.** *Let $Y$ be the mild solution to* (10) *and let $\mathrm{supp}(\mu) \subset H_\nu$. Then, $\mu_\varepsilon \ll \nu$ for any $\varepsilon > 0$.*

## 7. Applications

### 7.1. Gaussian mixture

To validate our methodology on a tractable example, we consider synthetic data sampled from the Gaussian mixture

$$\mu = \alpha\mathcal{N}(u, C) + (1 - \alpha)\mathcal{N}(-u, C) \tag{11}$$

on $H = L^2([0,1])$. We take a Matérn covariance operator $C$, which diagonalises with respect to the basis $\{\xi \mapsto \sin(j\xi) : j \ge 1\}$ on $H$ with eigenvalues

$$c_j = \sigma_0^2 \left( \rho_0^{-2} + (2\pi j)^2 \right)^{-1/2 + \nu_0},$$

see e.g. (Borovitskiy et al., 2020). We set $\sigma_0^2 = 10^3, \rho_0 = 5 \times 10^{-3}, \nu_0 = 1$ and $\alpha = 0.1$. The mean $u(\xi) = 2\,\xi^{1.5}(\pi - \xi)^{1.5}$ is chosen such that $u \in H_\nu$ for $\nu = \mathcal{N}(0, C)$. From the Cameron-Martin theorem it then follows that $\mu \ll \nu$. Samples of $\mu$ are displayed in Figure 3 (Appendix F).

In this setup, the true steering function $s(t, x)$ can be derived in closed form based on standard Gaussian computations. We examine our models performance under three distinct aspects; (i) its robustness to state dimension $D$, (ii) its robustness to small noising time $T$ as well as (iii) the impact of time-dependent noise schedules $\beta(t)$. For this, we implement the *forced* VP-SDE with $\gamma = 1$ and once with *constant noise (FCN)* and once with a linear *noise schedule (FNS)*, targeting (11) for various approximation dimensions $D$ and two noising times $T = 1$ and $T = 0.2$. For the approximation scheme, the state space $H$ is discretised in the spectral domain defined by the eigenbasis of the covariance operator $C$. As a baseline comparison, we implement the *noising-denoising model (ND)* as introduced in (Song et al., 2021) for finite-dimensional models as well as the *Hilbert space noising-denoising model (HND)* defined in (Pidstrigach et al., 2024). All models are implemented with the same neural network architecture, optimisation procedure and sampling step size.

*Table 1.* Averaged sliced Wasserstein distances between the true target and samples generated by the various diffusion models in the case $D = 500$, based on $N = 5000$ samples.

|  | ND | HND | FCN (ours) | FNS (ours) |
|---|---|---|---|---|
| $T = 1$ | 0.274 | 0.117 | 0.258 | 0.055 |
| $T = 0.2$ | 0.692 | 0.51 | 0.123 | **0.053** |

Figure 4 (Appendix F) shows the average distance $\mathbb{E}[\|s_\theta(t, X_t) - s(t, X_t)\|]$ between the score/steering approximation $s_\theta(t, x)$ and true score/steering term $s(t, x)$ during training for various state dimensions $D$ and $T = 1$. As expected, the models constructed to align with the infinite-dimensional geometry of the target measure perform independently of the dimension $D$, whereas the Euclidean diffusion model diverges as $D$ increases.

Samples obtained from the diffusion models for $D = 500$ are shown in Figure 5 (Appendix F) and to be compared to the true data samples in Figure 3 (Appendix F). The divergence of the ND model in the first row is apparent. Likewise, so is the collapse of the HND model in the limited noising time $T = 0.2$. In contrast, both forced models FCN and FNS stay robust to the decrease in time horizon. Quantitatively, this is supported by the averaged sliced Wasserstein distances displayed in Table 1.

We conclude that; (i) only the finite-dimensional method diverges as $D$ increases, (ii) only our forced diffusion models FCN and FNS remain robust to a decrease in noising time $T$ and (iii) the forced noise scheduled diffusion (FNS) consistently outperforms the forced constant noise model (FCN), even in this simple synthetic example.

### 7.2. MNIST-SDF

The performance of the proposed method is demonstrated on the MNIST-SDF dataset (Sitzmann et al., 2020), derived from MNIST samples by applying a signed distance transform to the pixel based images. This transformation renders the data inherently function space valued and hence suitable for the setup at hand.

We train the VP-SPDE model to target the data measure at a resolution of $64 \times 64$ pixels, upsampled from the original $32 \times 32$ resolution, and compare our results with those reported for a number of generative models in (Lim et al., 2025). For the sake of comparability, we adopt the network architecture proposed in their work, which consists of a $2D$ Fourier Neural Operator-based U-net with three resolution levels and spectral residual blocks with time embeddings and positional encoding. For details, see (Lim et al., 2025), Appendix J.

As our reference measure $\mathcal{N}(0, C)$, we choose $C$ to be diagonal in the discrete cosine space, and let the first $32 \times$

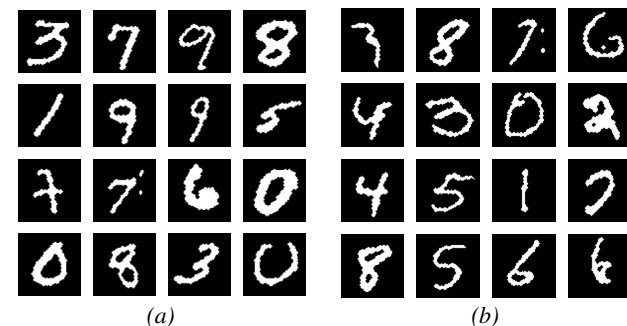

*Figure 1.* True and generated **MNIST-SDF** samples (masked) at $64 \times 64$ resolution. **(a):** True samples, upsampled to $64 \times 64$. **(b):** Generated samples returned by the VP-SPDE model.

32 entries correspond to the dataset's empirical marginal variances of each mode. For the remaining modes $j > 32$, we choose the smallest variance, scaled by a decay rate proportional to $j^{-(1+\varepsilon)}$ to ensure that $C$ is trace-class in the limit. In the VP-SPDE (7), we set $\gamma = 0.05$ and $\beta$ to follow a cosine noising schedule (Nichol & Dhariwal, 2021).

Figure 1 shows a comparison between true MNIST-SDF samples, upsampled to $64 \times 64$-resolution, and those generated by the VP-SPDE model. Figures are displayed after applying a binary mask, whereas Figure 6 (Appendix F) shows a comparison of the samples in SDF space. We additionally provide in Table 2 the Fréchet Inception Distance (FID) obtained by our model and put it in context with the results obtained by (Lim et al., 2025) for the Denoising-Diffusion-Operators (DDO) (Lim et al., 2025), GANO (Rahman et al., 2022) and Multilevel Diffusion model (MultiDiff) (Hagemann et al., 2025). While the FID score obtained by our method is competitive with the one obtained for DDO, it does not fully match its reported performance. We suspect that this is due to a lack of extensive hyperparameter tuning during training, necessitated by limited computational resources.

### 7.3. Bayesian inverse problem: seismic imaging

Consider a Bayesian inverse problem of inferring an unknown random variable $X$, given some prior measure $\mu_0$ and observations $Y$ generated by a forward model $Y = G(X) + \eta$. Here $G$ denotes an observation operator and $\eta$ observational noise. This setting defines a Bayesian inverse problem, in which the objective is to sample from the posterior distribution of $X$ given an observation $Y = y$.

*Table 2.* FID scores on MNIST-SDF. The values marked with $^\star$ have been reported in (Lim et al., 2025).

|  | GANO | MultiDiff | DDO | VP-SPDE (ours) |
|---|---|---|---|---|
| FID | 3.41$^\star$ | 35.09$^\star$ | **2.74**$^\star$ | 3.12 |

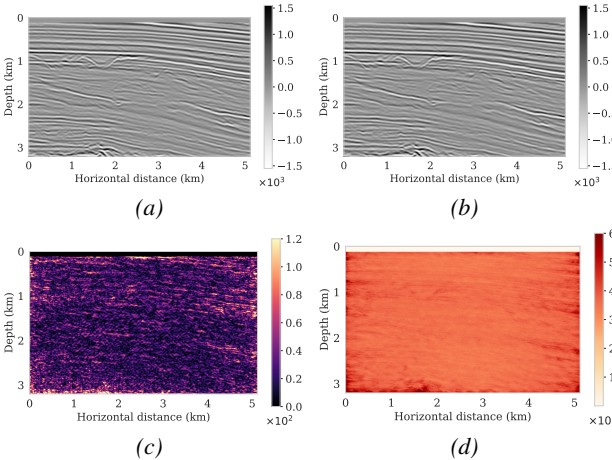

*Figure 2.* **Seismic imaging estimate** generated by the forced VP-SPDE. **(a)** Ground truth seismic image $x$. **(b)** Estimated posterior mean $\hat{x}$. **(c)** Absolute error between ground truth and posterior mean. **(d)** Estimated marginal posterior standard deviation.

Our methodology readily generalises to this setting by learning a conditional $h$-transform, defined by substituting the target $\mu$ in (3) by the posterior of $X \mid Y = y$. The functions $h(t, x, y)$ and $s(t, x, y)$ then depend on the observed variable $y$, which is to be encoded in the approximation $s_\theta(t, x, y)$. Given samples $(x_1, y_1), \ldots, (x_n, y_n)$ of the joint distribution of $(X, Y)$, training of $s_\theta(t, x, y)$ proceeds as before by minimising the loss (6). After training, at inference time, $y$ serves as an input parameter to generate samples of $X \mid Y = y$ by solving the forced diffusion model (4). This is similar to the approach of (Pidstrigach et al., 2024; Baldassari et al., 2023) of learning conditional time reversals.

We follow an inverse problem laid out in (Baldassari et al., 2023) of estimating the high-frequency imaging components $X$ of the Earth's subsurface squared-slowness model (Figure 9 (a), Appendix F), conditioned on a known, smooth long-wavelength background model (Figure 9 (b), Appendix F) and surface seismic observations. Here, following (Orozco et al., 2023), the high dimensionality of the surface data is reduced by applying the adjoint Born scattering operator to the measurements. The resulting transformed representation $Y$ (Figure 9 (c), Appendix F) is then used as conditioning variable in the generative diffusion model. A linearisation of the Born operator turns the problem into an inverse problem with linear operator $G$, albeit with non-trivial null space, see (Baldassari et al., 2023) for details.

We train the VP-SPDE with a linear noise schedule and white noise covariance operator. To evaluate the trained model, the posterior mean and marginal standard deviations are estimated using 1024 generated samples drawn from $X \mid Y = y$, where $y$ is a measurement not seen during training. Figure 2 shows the true target seismic image $x$, estimated

posterior mean $\hat{x}$ and marginal standard deviation as well as the absolute error between the target and posterior mean. For a baseline comparison, we additionally implement the noising-denoising model introduced in (Baldassari et al., 2023), whose results are presented in Fig. 10 (Appendix F).

Similar to the results reported in (Baldassari et al., 2023), the absolute error and marginal standard deviations obtained by either model are largest in areas of high geological complexity as well as near structural boundaries. However, in comparison to the noising-denoising baseline, our model exhibits improved generalisation and more accurately recovers complex geological structures.

This is quantitatively supported by the statistics in Table 3, with the VP-SPDE achieving lower relative and mean absolute errors than the baseline. Conversely, the baseline yields a lower average marginal standard deviation than the VP-SPDE. Note that the baseline's variance is notably small in well-estimated regions but increases significantly in areas of high reconstruction error. The baseline's average posterior standard deviation (24.27) is significantly smaller than it's average reconstruction error (37.8), indicating a problematic fit on the data. In contrast, our model exhibits a more uniform marginal variance across the domain, which, given the lower reconstruction error, suggests a more consistent representation of posterior uncertainty. We hypothesize that this is due to the forced VP-SPDE inducing a more efficient diffusion toward the target distribution, potentially reducing bias arising from the finite-time horizon and discretization effects.

## 8. Conclusion

In this paper, we have proposed a new method to construct infinite-dimensional generative diffusions by forcing SPDEs towards a target measure via Doob's $h$-transform. We have developed a rigorous mathematical theory showing the well-definedness of this method in infinite-dimensional spaces under appropriate assumptions, and also proven bounds on the Wasserstein-2 distance of the generated samples from the true distribution, showing asymptotic consistency. We can still use a computationally tractable score-matching loss function. Our method has the key advantage that the simulation time $T$ of the process can be kept relatively short, which would introduce significant bias in the time-reversed

*Table 3.* Quantitative results for the seismic imaging inverse problem, based on $N = 1024$ generated samples.

| Method | Relative error $\|\hat{x} - x\|/\|x\|$ | Avg. abs. error | Avg. poster. std. |
|---|---|---|---|
| VP-SPDE (ours) | **0.09** | **26.82** | 35.28 |
| Noising–denoising | 0.17 | 37.80 | 24.27 |

diffusion approaches. We illustrate the performance of our method on three examples (Gaussian-mixtures, MNIST-SDF and seismic imaging). It performs competitively with state-of-the-art alternatives both in terms of fidelity, as well as computational cost for training and inference.

In future work, it would be interesting to study the effect of the roughness parameter $\gamma$ and the choice of covariance operator $C$, as well as developing adaptive methods to choose the optimal time length $T$, step size $\Delta t$ and number of Langevin steps at initialisation. We believe that the performance of forced generative SPDE models can still improve significantly with further optimization of these parameters.

## Impact Statement

The primary objective of this work is to advance the theoretical foundations of generative diffusion models in infinite-dimensional settings. The results obtained in this paper may ultimately support applications in generative modeling and scientific computing. We do not foresee immediate negative societal or ethical consequences arising from this work beyond those already well established for generative modeling research.

## Acknowledgements

This research is supported by the Ministry of Education, Singapore, under its Academic Research Fund Tier 1 (RS19/25): Efficient sampling of infinite-dimensional conditioned diffusions. We would like to thank Giulio Franzese for insightful discussions.

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

# A. Related Work

## A.1. Generative models

Advancing generative diffusion models purely from the noising-denoising framework to diffusion bridges and optimal transport formulations has seen plenty of interest in the past years. In the foundational work by (De Bortoli et al., 2021), a generative diffusion is phrased as the solution to the (Diffusion) Schrödinger bridge (DSB) problem, enabling diffusion models to interpolate between arbitrary measures instead of relying on Gaussian priors. The DSB problem is solved by iteratively updating forward and backward diffusion dynamics to alternately match the initial and terminal distributions, hence relying on the noising-denoising score matching framework. In (Shi et al., 2023; Tong et al., 2023), alternative, simulation-free DSB matching objectives have been proposed to circumvent the heavy computational burden of simulating full diffusion paths forward and backward in time. In (Chen et al., 2021), the DSB problem has been studied under the framework of stochastic optimal control (SOC), and the methodology presented there avoids iterative projections by directly targeting the underlying Hamilton-Jacobi-Bellman (HJB) equations via Feynman-Kac representations of the optimal control policy.

In parallel to the DSB formulation, alternative diffusion model frameworks have emerged that circumvent the reliance on time reversals entirely. In (Peluchetti, 2023b) the diffusion bridge mixture transport (DBMT) is introduced, a non-denoising framework that constructs generative models forward in time using a mixture of diffusion bridges. Compared to the DSB problem, it solves a simpler problem, but circumvents the need for an iterative training procedure of denoising diffusions and expensive forward-backward path simulations. In a complementary approach, (Liu et al., 2022) utilizes diffusion bridges to unify the generative learning process under a single KL divergence objective, extending latent variable diffusions to non-Euclidean and discrete spaces. Furthermore, (Peluchetti, 2023a) establishes that iterating the DBMT procedure recovers the exact Schrödinger bridge, achieving optimal transport without time reversals. In (Christensen et al., 2025), a unified framework for generative modeling is developed using Doob's h-transforms, generalizing the bridge construction beyond fixed time horizons.

To the best of our knowledge, the only comparable result that extends the rich literature on generative diffusion bridges to the infinite-dimensional setting is given in (Park et al., 2024). Here, Ornstein-Uhlenbeck bridges - a linear diffusion pinned on both an initial and terminal state - are derived from a SOC perspective and subsequently used to extend the DBMT approach of (Peluchetti, 2023b). The methodology is restricted to time autonomous, diagonalisable diffusions. Moreover, no existence results for the diffusion bridge mixture are given.

In contrast, our work provides an alternative probabilistic framework that defines a generative diffusion, forced to sample exactly from a target distribution $\mu$, based on one unified change of measure $\mathbb{P}^h$. This change of measure perspective is particularly useful in the infinite-dimensional setting, as it can be understood as a Girsanov-type transformation under which existence of mild solutions to SPDEs are well understood in the existing literature (Da Prato & Zabczyk, 2014; Pieper-Sethmacher et al., 2025a). In practice, this offers a wider flexibility in the choice of the reference diffusion $X$ compared to (Peluchetti, 2023b; Park et al., 2024), allowing the forcing of non-autonomous or even non-linear diffusions and SPDEs with unbounded drift operators. As an additional consequence, existence results of the forced diffusion as a mild solution to an infinite-dimensional equation can be stated under verifiable assumptions on the target $\mu$ and the reference process $X$. This allows us to obtain 2-Wasserstein bounds between the target measure and the learned measure.

Our current framework prioritizes the construction of a generative diffusion model based on a learnable change of measure. The literature on finite-dimensional models suggests a rich connection between our approach and the SOC and DSB perspectives. This represents a promising future research direction, in particular in solving an infinite-dimensional DSB without time-reversals as well as connecting our $h$-function of choice to the solution of a stochastic optimal control problem.

## A.2. Diffusion bridges

There has also been a recent interest in sampling *infinite-dimensional diffusion bridges* (Baker et al., 2024; Yang et al., 2024; Pieper-Sethmacher et al., 2025b). While the terminology implies similarities, simulating diffusion bridges and learning generative diffusion models are fundamentally different problems. In the former, a *fixed*, typically non-linear diffusion process of interest is conditioned to hit an observed terminal state at some time $T > 0$. In the case of linear dynamics, the diffusion bridge can be derived analytically by what is known as the Brownian ($A = 0, F = 0$) or Ornstein-Uhlenbeck bridge ($F = 0$). If the dynamics are non-linear, the problem becomes more difficult since the conditioned bridge process - derived by an application of Doob's $h$-transform - includes a steering term based on the intractable transition densities of the

non-linear diffusion process. In (Baker et al., 2024), a score-matching approach in the spirit of (Heng et al., 2025) is applied to approximate the steering term by learning two time-reversals. This is extended in (Yang et al., 2024), which includes an operator learning framework to learn the steering term directly in function space, independently of any specific discretisation. In (Pieper-Sethmacher et al., 2025b), a Hilbert space valued Markov chain Monte Carlo scheme was introduced that draws exact samples from an infinite-dimensional diffusion bridge.

In contrast, the work at hand concerns *generative modeling*, in which one aims to train a sampler that targets an unknown distribution $\mu$, given iid samples $y_1, \ldots y_N$ thereof. Except in the trivial case that $\mu$ is a Dirac measure, this is a fundamentally different problem. Our proposed method addresses this problem by conditioning a reference diffusion to satisfy $\mu$ at time $T$. While the conditioning is also based on Doob's $h$-transform, the intractability of the resulting steering term stems from the unknown target measure $\mu$, not from the complexity of the reference diffusion process. In fact, our method allows for a free choice of the reference diffusion, and it is typically beneficial to choose a linear reference to reduce sampling and training costs.

## B. Proofs for Section 3

We give here the detailed proof for Theorem 3.3. This can be divided up into three separate steps;

1. Show that $h$ defines a change of measure $\mathbb{P}^h$.

2. Show that, under the changed measure $\mathbb{P}^h$, the process $X$ at time $T$ has the desired distribution $\mu$.

3. Show that, under $\mathbb{P}^h$, $X$ satisfies the diffusion equation given in (4).

The first step is covered in the following two lemmas.

**Lemma B.1.** *The function $h$ as defined in Equation (3) is space-time harmonic in the sense that*

$$\mathbb{E}[h(t+s, X_{t+s}) \mid X_s = x] = h(s, x), \quad s + t < T.$$

*Proof.* A simple computation shows that

$$
\begin{aligned}
\mathbb{E}[h(t+s, X_{t+s}) \mid X_s = x] &= \int_H h(t+s, z) p(s, x; s+t, z)\, \nu(\mathrm{d}z) \\
&= \int_H \int_H p(t+s, z; T, y) p(s, x; s+t, z) \frac{1}{p_T(y)}\, \mathrm{d}\mu(y)\, \nu(\mathrm{d}z) \\
&= \int_H p(s, x; T, y) \frac{1}{p_T(y)}\, \mathrm{d}\mu(y) \\
&= h(s, x).
\end{aligned}
$$

Here, we have used Assumption 3.1 in the first step, the definition of $h$ in the second step and the Chapman-Kolmogorov equation in step three. $\square$

**Lemma B.2.** *The process $h(t, X_t), t < T$, is a $\mathbb{P}$-martingale with $\mathbb{E}[h(t, X_t)] = 1$. In particular, there exists a unique change of measure $\mathbb{P}^h$ on $\mathcal{F}_T$ defined by*

$$\mathrm{d}\mathbb{P}_t^h = h(t, X_t)\, \mathrm{d}\mathbb{P}_t^h, \quad t < T,$$

*where $\mathbb{P}_t^h, \mathbb{P}_t$ denote the restrictions of $\mathbb{P}^h$ and $\mathbb{P}$ onto $\mathcal{F}_t$.*

*Proof.* From Lemma B.1 and the Markov property of $X$ it follows that

$$\mathbb{E}[h(t, X_t) \mid \mathcal{F}_s] = \mathbb{E}[h(t, X_t) \mid X_s] = h(s, X_s),$$

which shows the martingale property of $h(t, X_t)$. Moreover,

$$\mathbb{E}[h(0, X_0)] = \int_H \int_H \frac{p(0, x_0; T, y)}{p_T(y)}\, \mathrm{d}\mu_0(x_0)\, \mathrm{d}\mu(y) = \int_H \frac{p_T(y)}{p_T(y)}\, \mathrm{d}\mu(y) = 1.$$

The second claim then follows from the Carathéodory extension theorem, see e.g. (Stroock, 1987), Lemma 4.2. $\square$

Step Two is a consequence of the lemma below.

**Lemma B.3.** *Under the measure $\mathbb{P}^h$ of Lemma B.2, it holds for any bounded and measurable function $g$ that*

$$\mathbb{E}^h[g(X_t)] = \int_H \mathbb{E}\left[g(X_t) \mid X_T = y\right] \mu(\mathrm{d}y). \tag{12}$$

*Proof.* An application of the Bayes theorem in the third step gives

$$\mathbb{E}^h\left[g(X_t)\right] = \mathbb{E}\left[g(X_t)h(t, X_t)\right] = \int_H \int_H g(x)\frac{p(t, x; T, y)}{p_T(y)}p_t(x)\,\mathrm{d}\mu(y)\,\mathrm{d}\nu(x) = \int_H \mathbb{E}[g(X_t) \mid X_T = y]\,\mu(\mathrm{d}y).$$

$\square$

Step three is proven in the following.

*Proof.* (remainder of Theorem 3.3)

Let $L$ denote the space-time generator of $(t, X_t)$ under $\mathbb{P}$, defined as

$$(L\varphi)(t, x) := \lim_{s \to t} \frac{\mathbb{E}[\varphi(t, X_t) \mid X_s = x] - \varphi(s, x)}{t - s}$$

for any $\varphi$ such that the limit exists in the topology of bounded pointwise convergence (Priola, 1999; Fabbri et al., 2017). Under the assumed Fréchet differentiability of $h$ in Assumption 3.1, it follows from (Pieper-Sethmacher et al., 2025a), Lemma 3.3. that

$$h(t, X_t) = h(0, X_0) + \int_0^t Lh(s, X_s)\,\mathrm{d}s + \int_0^t \langle B^*(s)\,\mathrm{D}_x h(s, X_s), \mathrm{d}W_s\rangle.$$

By the space-time harmonic property of $h$ in Lemma B.1, we have $Lh = 0$. Hence, defining $\bar{h}(t, X_t) = \frac{h(t, X_t)}{h(0, X_0)}$, it holds

$$\mathrm{d}\bar{h}(t, X_t) = \frac{1}{h(0, X_0)}\langle B^*(t)\,\mathrm{D}_x h(t, X_t), \mathrm{d}W_t\rangle$$
$$= \bar{h}(t, X_t)\langle B^*(t)\,\mathrm{D}_x \log h(t, X_t), \mathrm{d}W_t\rangle$$

with $\bar{h}(0, X_0) = 1$. It follows that $h(t, X_t) = h(0, X_0)\mathcal{E}(M^h)_t$, where $\mathcal{E}(M^h)$ denotes the Doléans-Dade exponential of the martingale

$$M_t^h := \int_0^t \langle B^*(s)\,\mathrm{D}_x \log h(s, X_s), \mathrm{d}W_s\rangle.$$

Given the martingale property of $h(t, X_t)$, the Girsanov theorem implies that the process

$$W_t^h = W_t - \int_0^t B^*(s)\,\mathrm{D}_x \log h(s, X_s)\,\mathrm{d}s$$

is a cylindrical Wiener process under the changed measure $\mathbb{P}^h$ defined by $\mathrm{d}\mathbb{P}_t^h = h(t, X_t)\,\mathrm{d}\mathbb{P}^h$ for any $t < T$. Plugging this into Equation (2) shows that $X$ under $\mathbb{P}^h$ satisfies the forced diffusion (4).

$\square$

## C. Proofs for Section 4

*Proof.* (of Proposition 4.1) From the Girsanov theorem, it follows that $\mathbb{X}^\theta$ and $\mathbb{X}^h$ are equivalent with Radon-Nikodym derivative

$$\frac{\mathrm{d}\mathbb{X}^\theta}{\mathrm{d}\mathbb{X}^h}(X) = \frac{\mathrm{d}\mu_0}{\mathrm{d}\mu_0^h}(X_0)\exp\left(\int_0^T \langle\eta_\theta(t), \mathrm{d}W_t^h\rangle\,\mathrm{d}t - \frac{1}{2}\int_0^T |\eta_\theta(t)|^2\,\mathrm{d}t\right),$$

where $W^h$ is a cylindrical Wiener process and $\eta_\theta(t) = B^*(t)(s_\theta(t, X_t) - s(t, X_t))$. Since the stochastic integral term vanishes under expectation with respect to $\mathbb{P}^h$, it follows that

$$D_{\mathrm{KL}}(\mathbb{X}^h \| \mathbb{X}^\theta) = -\mathbb{E}^h \left[ \log \left( \frac{\mathrm{d}\mathbb{X}^\theta}{\mathrm{d}\mathbb{X}^h} \right) \right]$$

$$= -\mathbb{E}^h \left[ \log \frac{\mathrm{d}\mu_0}{\mathrm{d}\mu_0^h}(X_0) \right] + \frac{1}{2} \mathbb{E}^h \left[ \int_0^T |\eta_\theta(t)|^2 \, \mathrm{d}t \right].$$

Plugging in the definition of $\eta_\theta(t)$, the second term on the right-hand side of equals

$$\frac{1}{2} \mathbb{E}^h \left[ \int_0^T |B^*(t)(s_\theta(t, X_t) - s(t, X_t))|^2 \, \mathrm{d}t \right] = \frac{1}{2} \mathbb{E}^h \left[ \int_0^T |B^*(t)s(t, X_t)|^2 + |B^*(t)s_\theta(t, X_t)|^2 \right.$$

$$\left. - 2 \langle B^*(t)s(t, X_t), B^*(t)s_\theta(t, X_t) \rangle \, \mathrm{d}t \right]. \tag{13}$$

Notice that the first term in the integral on the right-hand side does not depend on $\theta$ and can hence be treated as a constant while minimising in $\theta$. Moreover, denoting by $\langle x, y \rangle_{B_t^*} = \langle B^*(t)x, B^*(t)y \rangle$, it holds that

$$
\begin{aligned}
\mathbb{E}^h \left[ \langle s(t, X_t), s_\theta(t, X_t) \rangle_{B_t^*} \right] &= \mathbb{E}^h \left[ \langle \mathrm{D}_x \log h(t, X_t), s_\theta(t, X_t) \rangle_{B_t^*} \right] \\
&= \mathbb{E} \left[ \langle \mathrm{D}_x \log h(t, X_t), s_\theta(t, X_t) \rangle_{B_t^*} h(t, X_t) \right] \\
&= \mathbb{E} \left[ \langle \mathrm{D}_x h(t, X_t), s_\theta(t, X_t) \rangle_{B_t^*} \right] \\
&= \int_H \langle \mathrm{D}_x h(t, x), s_\theta(t, x) \rangle_{B_t^*} p_t(x) \, \nu(\mathrm{d}x) \\
&= \int_H \left\langle \mathrm{D}_x \left( \int_H \frac{p(t, x; T, y)}{p_T(y)} \mu(\mathrm{d}y) \right), s_\theta(t, x) \right\rangle_{B_t^*} p_t(x) \, \nu(\mathrm{d}x) \\
&= \int_H \int_H \langle \mathrm{D}_x \log p(t, x; T, y), s_\theta(t, x) \rangle_{B_t^*} \frac{p(t, x; T, y)}{p_T(y)} p_t(x) \, \nu(\mathrm{d}x) \, \mu(\mathrm{d}y) \\
&= \int_H \mathbb{E} \left[ \langle \mathrm{D}_{x_t} \log p(t, X_t; T, y), s_\theta(t, X_t) \rangle_{B_t^*} \mid X_T = y \right] \mu(\mathrm{d}y) \\
&= \mathbb{E}^h \left[ \langle \mathrm{D}_{x_t} \log p(t, X_t; T, X_T), s_\theta(t, X_t) \rangle_{B_t^*} \right].
\end{aligned}
\tag{14}
$$

Here, we used the definition of $h$ and $\mathbb{P}^h$ in the first three steps, Assumption 3.1 in the fourth step, Fubini's theorem in step six and Equation (12) in the last step. Hence, in total

$$D_{\mathrm{KL}}(\mathbb{X}^h \| \mathbb{X}^\theta) = C_1 + C_2 + \frac{1}{2} \int_0^T \mathbb{E}^h \left[ |B^*(t)s_\theta(t, X_t)|^2 \right] \mathrm{d}t$$

$$- \int_0^T \mathbb{E}^h \left[ \langle B^*(t) \mathrm{D}_{x_t} \log p(t, X_t; T, X_T), B^*(t)s_\theta(t, X_t) \rangle \right] \mathrm{d}t$$

with $C_1 = -\mathbb{E}^h \left[ \log \frac{\mathrm{d}\mu_0}{\mathrm{d}\mu_0^h}(X_0) \right]$ and $C_2 = \frac{1}{2} \mathbb{E}^h \left[ \int_0^T |B^*(t)s(t, X_t)|^2 \, \mathrm{d}t \right]$.

On the other hand, it holds that

$$
\begin{aligned}
\mathcal{L}(\theta) &= \frac{1}{2} \int_0^T \mathbb{E}^h \left[ |B^*(t) \left( s_\theta(t, X_t) - \mathrm{D}_{x_t} \log p(t, X_t; T, X_T) \right) |^2 \right] \mathrm{d}t \\
&= C_3 + \frac{1}{2} \int_0^T \mathbb{E}^h \left[ |B^*(t)s_\theta(t, X_t)|^2 \right] \mathrm{d}t - \int_0^T \mathbb{E}^h \left[ \langle B^*(t)s_\theta(t, X_t), B^*(t) \mathrm{D}_{x_t} \log p(t, X_t; T, X_T) \rangle \right] \mathrm{d}t
\end{aligned}
\tag{15}
$$

with $C_3 = \frac{1}{2} \int_0^T \mathbb{E}^h \left[ |B^*(t) \mathrm{D}_{x_t} \log p(t, X_t; T, X_T)|^2 \right] \mathrm{d}t$. Hence, in total, $D_{\mathrm{KL}}(\mathbb{X}^h \| \mathbb{X}^\theta) = C_1 + C_2 - C_3 + \mathcal{L}(\theta)$ which proves the claim. $\qquad \square$

*Proof.* (of Corollary 4.4) Let $\theta^\star = \arg\min_{\theta \in \Theta} \mathcal{L}(\theta)$ and define

$$\tilde{\mathcal{L}}(\theta) = \frac{1}{2}\mathbb{E}^h \left[ \int_0^T |B^*(t)(s_\theta(t, X_t) - s(t, X_t))|^2 \, dt \right].$$

Following Equations (13) to (15), we then have $\theta^\star = \arg\min_{\theta \in \Theta} \tilde{\mathcal{L}}(\theta^\star)$. On the other hand, given Assumption 4.3, there exists some $\theta \in \Theta$ such that $\tilde{\mathcal{L}}(\theta) = 0$ and hence $\theta^\star$ satisfies $\tilde{\mathcal{L}}(\theta^\star) = 0$. Noting that $\ker(B^*(t)) = \{0\}$, it follows for almost every $t \in [0, T]$ that

$$|s_{\theta^\star}(t, X_t) - s(t, X_t)| = 0 \quad \mathbb{P}^h\text{-a.s.} \tag{16}$$

From the continuity of $s$ and a.s. continuity of $t \mapsto X_t$, one concludes that (16) holds uniformly for all $t \in [0, T]$. In particular, we have $s_{\theta^\star}(0, x) = s(0, x)$ for $\mu_0$-a.e. $x \in H$, and hence $X_0^{\theta^\star} = X_0^h$ in law. Jointly with $\tilde{\mathcal{L}}(\theta^\star) = 0$, this gives that $D_{\mathrm{KL}}(\mathbb{X}^h \| \mathbb{X}^{\theta^\star}) = 0$.

On the other hand, let $\theta^\star$ be such that $\mathbb{X}^{\theta^\star} = \mathbb{X}^h$. Then also $D_{\mathrm{KL}}(\mathbb{X}^h \| \mathbb{X}^{\theta^\star}) = 0$ and hence $\tilde{\mathcal{L}}(\theta^\star) = 0$. Again, by Equations (13) to (15), we then have $\theta^\star = \arg\min_{\theta \in \Theta} \mathcal{L}(\theta^\star)$.

$\square$

# D. Proofs for Section 5

*Proof.* (of Proposition 5.1) Fix some $\gamma > 0$. Since $C$ is a positive, trace-class operator on $H$, there exists an orthonormal basis $(e_j)_j$ of $H$ and a sequence of positive eigenvalues $c_j$ such that $\sum_{j=1}^\infty c_j < \infty$ and $Ce_j = c_j e_j$ for all $j \geq 1$. Hence, the inverse $C^{-\gamma}$ is an unbounded operator on $H$ with dense domain

$$\mathcal{D}_{C^{-\gamma}} = \{x \in H : \sum_{j \geq 1} c_j^{-2\gamma} |\langle x, e_j \rangle|^2 < \infty\}.$$

Let $B(t) = \sqrt{\beta(t)Q}$ and $A(t) = -\frac{1}{2}\beta(t)C^{-\gamma}$. The operator family $A = (A(t))_{t \geq 0}$ is then densely defined with common domain $\mathcal{D}_A = \mathcal{D}_{C^{-\gamma}}$ and generates an evolution family of diagonalisable operator $U(t, s)$ defined by

$$U(t, s)e_j = \exp\left(-\alpha_j \int_s^t \beta(r) \, dr\right) e_j \tag{17}$$

where $\alpha_j = \frac{1}{2}c_j^{-\gamma}$. A straightforward computation then shows that

$$\int_0^T \mathrm{tr}\left[U(t, 0)B(t)B^*(t)U^*(t, 0)\right] dt < \infty$$

for any $T > 0$. From this it follows (Da Prato & Zabczyk, 2014; Knäble, 2011) that Equation (7) admits a unique mild solution for any initial value $X_s = x \in H, s > 0$, given by

$$X_t = U(t, s)x + \int_s^t U(t, r)B(r) \, dW_r, \quad s \leq t \leq T.$$

The random variable $X_t \mid X_s = x$ is Gaussian on $H$ with mean $U(t, s)x$ and covariance operator $Q(t, s) = \int_s^t U(t, r)B(r)B^*(r)U^*(t, r) \, dr$, which remains diagonal with eigenvalues

$$q_j(t, s) = \frac{q_j}{2a_j}\left[1 - \exp\left(-2a_j \int_s^t \beta(r) \, dr\right)\right],$$

where $q_j = c_j^{1-\gamma}$. We are left to verify that $\mathcal{N}(U(t, s)x, Q(t, s))$ is absolutely continuous with respect to $\mathcal{N}(0, C)$ for all $s \leq t \leq T$ and $x \in H$. We first show that the measures $\{\mathcal{N}(U(t, s)x, Q(t, s)) : x \in H\}$ are equivalent. The eigenvalue expansions of $U(t, s)$ and $Q(t, s)$ show that $(Q(t, s))^{-\frac{1}{2}}U(t, s)$ has eigenvalues

$$\frac{\sqrt{2a_j} \exp\left(-\alpha_j \int_s^t \beta(r) \, dr\right)}{\sqrt{q_j\left[1 - \exp\left(-2a_j \int_s^t \beta(r) \, dr\right)\right]}},$$

which, since $a_j \to \infty$ sufficiently fast, are bounded in $j \geq 1$ for any fixed $s \leq t$. Hence $(Q(t,s))^{-\frac{1}{2}}U(t,s)$ is a well-defined, bounded operator and $\operatorname{im} U(t,s) \subset \operatorname{im} Q(t,s)^{\frac{1}{2}}$. From the Cameron-Martin theorem it follows that $\{\mathcal{N}(U(t,s)x, Q(t,s)) : x \in H\}$ are equivalent.

It remains to show that $\mathcal{N}(0, Q(t,s))$ is absolutely continuous with respect to $\mathcal{N}(0,C)$. Another eigenvalue expansion shows that $C^{-\frac{1}{2}}(Q(t,s))^{\frac{1}{2}}$ and $(Q(t,s))^{-\frac{1}{2}}C^{\frac{1}{2}}$ are well-defined and bounded operators and hence $\operatorname{im} C^{\frac{1}{2}} = \operatorname{im}(Q(t,s))^{\frac{1}{2}}$. Moreover, the operator $(C^{-\frac{1}{2}}Q(t,s)C^{-\frac{1}{2}} - I)$ has eigenvalues $(-\exp(-2a_j \int_s^t \beta(r)\,\mathrm{d}r))$ and is thus Hilbert-Schmidt. The conclusion then follows from the Feldman-Hajek theorem.

Lastly, the Fréchet differentiability of $h(t,x)$ for any $t < T$ follows from the differentiability of the Ornstein-Uhlenbeck transition densities of $\mathcal{N}(U(t,s)x, Q(t,s))$ with respect to $\mathcal{N}(0,C)$ for any $s < t, x \in H$, see for example (Pieper-Sethmacher et al., 2025a), Section 4.2.

$\qquad\square$

# E. Proofs for Section 6.1

*Proof of Proposition 6.1.* The proof of this result closely follows the lines of the proof of Theorem 14 of (Pidstrigach et al., 2024).

We start by coupling a process $\hat{X}_t^h$ following (8) initiated in initial distribution $\hat{X}_0^h \sim \hat{\mu}_0^h$ with another process $\tilde{X}_t^h$ following (8) with $s_\theta(x,t)$ instead of $s(x,t)$, initiated in $\tilde{X}_0^h \sim \mu_0^h$.

In both cases, we can use the unique mild solution similarly to (2), with $U(t,s)$ generated by $A(t) = \beta(t)A$, and $F(s, X_s)$ replaced by the Doob's $h$ transform forcing term $\beta(s)Qs(t, X_s^h)$,

$$
\begin{aligned}
X_t^h &= U(t,0)X_0^h + \int_0^t U(t,s)\beta(s)Qs(t, X_s^h)\,\mathrm{d}s + \int_0^t U(t,s)\sqrt{\beta(s)Q}\,\mathrm{d}W_s, \\
\tilde{X}_t^h &= U(t,0)\hat{X}_0^h + \int_0^t U(t,s)\beta(s)Qs_\theta(t, \tilde{X}_s^h)\,\mathrm{d}s + \int_0^t U(t,s)\sqrt{\beta(s)Q}\,\mathrm{d}W_s
\end{aligned}
\tag{18}
$$

Using the diagonal form of $U(t,s)$ from (17), it follows that $\|U(t,s)e_j\| \leq 1$ for any $0 \leq s \leq t$.

For the first term, we can show by the definition of the loss function (6) that

$$
\mathbb{E}_{\mathbb{P}_h}\left[\left(\int_0^t U(t,s)\beta(s)Q(s(t, X_s^h) - s_\theta(t, X_s^h))\,\mathrm{d}s\right)^2\right] \leq \mathcal{L}(\theta) \text{ for } 0 \leq t \leq T.
$$

Following a similar argument to the proof of Theorem 2 of (Chen et al., 2022), it follows that

$$
\mathbb{E}\|X_T^h - \tilde{X}_T^h\|^2 \leq C\exp(2LT)\mathcal{L}(\theta),
$$

for some constant $C$ independent of $T$. From this, we obtain that for the distribution $\tilde{X}_T^h \sim \tilde{\mu}_T^h$, we have

$$
\mathcal{W}_2(\tilde{\mu}_T^h, \mu) \leq \sqrt{C\mathcal{L}(\theta)}\exp(LT)
\tag{19}
$$

Now we move on a third process called $\hat{X}_t^h$, which follows (4) with estimated score $s_\theta(t,x)$, similarly to $\tilde{X}_t^h$, but we initiate it at the feasible initial distribution $\hat{X}_0^h \sim \hat{\mu}_0^h$ (via Langevin steps). We synchronously couple this with $\tilde{X}_t^h$, their unique mild solutions can be written as

$$
\begin{aligned}
\tilde{X}_t^h &= U(t,0)X_0^h + \int_0^t U(t,s)\beta(s)Qs_\theta(t, X_s^h)\,\mathrm{d}s + \int_0^t U(t,s)\sqrt{\beta(s)Q}\,\mathrm{d}W_s, \\
\hat{X}_t^h &= U(t,0)\hat{X}_0^h + \int_0^t U(t,s)\beta(s)Qs_\theta(t, \hat{X}_s^h)\,\mathrm{d}s + \int_0^t U(t,s)\sqrt{\beta(s)Q}\,\mathrm{d}W_s.
\end{aligned}
\tag{20}
$$

If we couple these with the same noise $W_s$, we can see that the difference evolves as

$$
X_t^h - \hat{X}_t^h = U(t,0)(X_0^h - \hat{X}_0^h) + \int_0^t U(t,s)\beta(s)Q(s(t, X_s^h) - s(t, \hat{X}_s^h))\,\mathrm{d}s.
\tag{21}
$$

---

**Algorithm 1** Training Forced Generative VP-SPDE

---

**Require:** Training data $y_1, \ldots, y_N$, covariance operator $C$, initial model parameters $\theta$, batch size $B$
1: **while** not converged **do**
2:  Sample $y_1, \ldots, y_B$ from data, $t_1, \ldots, t_B \sim \mathcal{U}(0, T)$ and Ornstein-Uhlenbeck bridges $X_{t_1}^{y_1}, \ldots, X_{t_B}^{y_B} \sim \mathbb{P}^h$
3:  Approximate loss $\mathcal{L}(\theta)$ in (6) via

$$\hat{\mathcal{L}}(\theta) = \frac{T}{B} \sum_{i=1}^{B} \left| s_\theta(t_i, X_{t_i}^{y_i}) - \mathrm{D}_x \log p(t_i, X_{t_i}^{y_i}; T, y_i) \right|^2_{B_{t_i}^*}$$

4:  Update model parameters $\theta$ via stochastic gradient descent using $\nabla_\theta \hat{\mathcal{L}}(\theta)$
5: **end while**
6: **return** Approximated steering function $s_\theta$

---

By Grönwall's lemma, using the Lipschitz assumption that $\|\beta(s)Q(s(t, X_s^h) - s(t, \hat{X}_s^h))\| \leq L\|X_s^h - \hat{X}_s^h\|$, it follows that for this synchronous coupling,

$$\|X_T^h - \hat{X}_T^h\| \leq \exp(LT)\|X_0^h - \hat{X}_0^h\|.$$

By the definition of the 2-Wasserstein distance, and the fact $X_T^h$ is distributed as the target $\mu$, this implies that

$$\mathcal{W}_2(\hat{\mu}_T^h, \mu) \leq \exp(LT)\mathcal{W}_2(\hat{\mu}_0^h, \mu_0^h), \tag{22}$$

with $\hat{\mu}_T^h$ denoting the distribution of $\hat{X}_T^h$. The size of the term $\mathcal{W}_2(\hat{\mu}_0^h, \mu_0^h)$ can be reduced by doing further MCMC steps at initialisation.

Finally, for the numerical discretization error for the process $\hat{X}_T^t$ using estimated score $s_\theta(t, x)$, we can use Theorem 3.14 of (Kruse, 2014), implying a bound of the form $O(\Delta t + \Delta x)$. The claim follows by combining this with bounds (19) and (22) via the triangle inequality. $\square$

*Proof.* (of Lemma 6.4) For any $t > 0$, the mild solution $Y_t$ given $Y_0 = y_0$ is Gaussian with mean $S_t y_0$ and covariance operator $Q_t = \int_0^t S_r C^{1-\gamma} S_r^* \, \mathrm{d}r$, where $S_t = \exp(-\frac{1}{2}C^{-\gamma}t)$ is the semigroup generated by $-\frac{1}{2}C^{-\gamma}$.

Just as in the proof of Proposition 5.1, an eigenvalue decompositions shows that $\nu = \mathcal{N}(0, C)$ and $\mathcal{N}(0, Q_t)$ are equivalent Gaussian measures. In particular, they share the same Cameron-Martin space $H_\nu$. Hence, by the Cameron-Martin theorem and the fact that $y_0 \in H_\nu$, it follows that $\mathcal{N}(S_t y_0, Q_t) \ll \mathcal{N}(0, C)$. Marginalising $Y_t$ by integrating out $Y_0 \sim \mu$ with $\mathrm{supp}(\mu) \subset H_\nu$ then finishes the proof.

$\square$

# F. Supplementary material for Section 7

### F.1. Algorithmic summary

The training and sampling procedure of the forced generative VP-SPDE are summarised in Algorithm 1 and 2 respectively. During training, the convergence criteria are user-defined and can be based on, for example, empirical metrics, the loss function or a fixed number of training steps.

For the time integration step in Algorithm 2, the numerical SPDE solver can be chosen by the user. In our experiments, we take a semi-implicit Euler-Maruyama scheme. One Langevin sampling step is effectively as expensive as one time integration step of the forced diffusion model. The number of Langevin steps should therefore be carefully weighed against the number of time integration steps at sample generation time. A principled way to determine the required Langevin steps is to initialise multiple parallel chains at $\mu_0$ and compute the Gelman-Rubin diagnostics $\hat{R}$ as a function of the number of steps (as test function, we can use the potential of $\mu_0$, for example). Typically $\hat{R} < 1.05$ indicates that convergence to the stationary distribution has occurred. In our examples, due to sufficient noising time, we have found the necessity of including Langevin steps only in the Gaussian mixture experiment when setting $T = 0.2$.

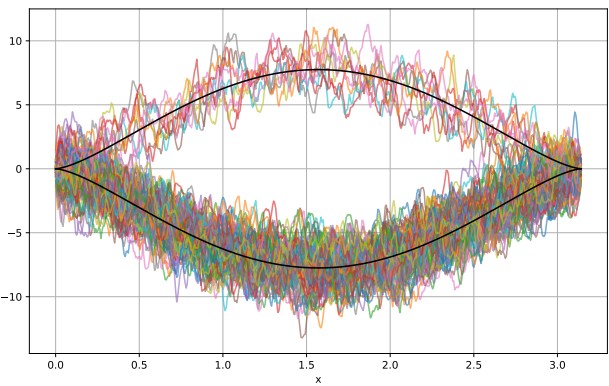

*Figure 3.* **Gaussian mixture** samples from the true target distribution defined in (11).

*Figure 4.* Distances $\mathbb{E}[\|s_\theta(t, X_t) - s(t, X_t)\|^2]$ between the true score/steering functions $s(t, x)$ and their approximations $s_\theta(t, x)$ for the various generative diffusion models and various state dimensions $D$. **(a):** the finite-dimensional denoising SDE. **(b):** the infinite-dimensional denoising SDE. **(c):** our forced VP-SPDE with constant noise. **(c):** our forced VP-SPDE with noise scheduling.

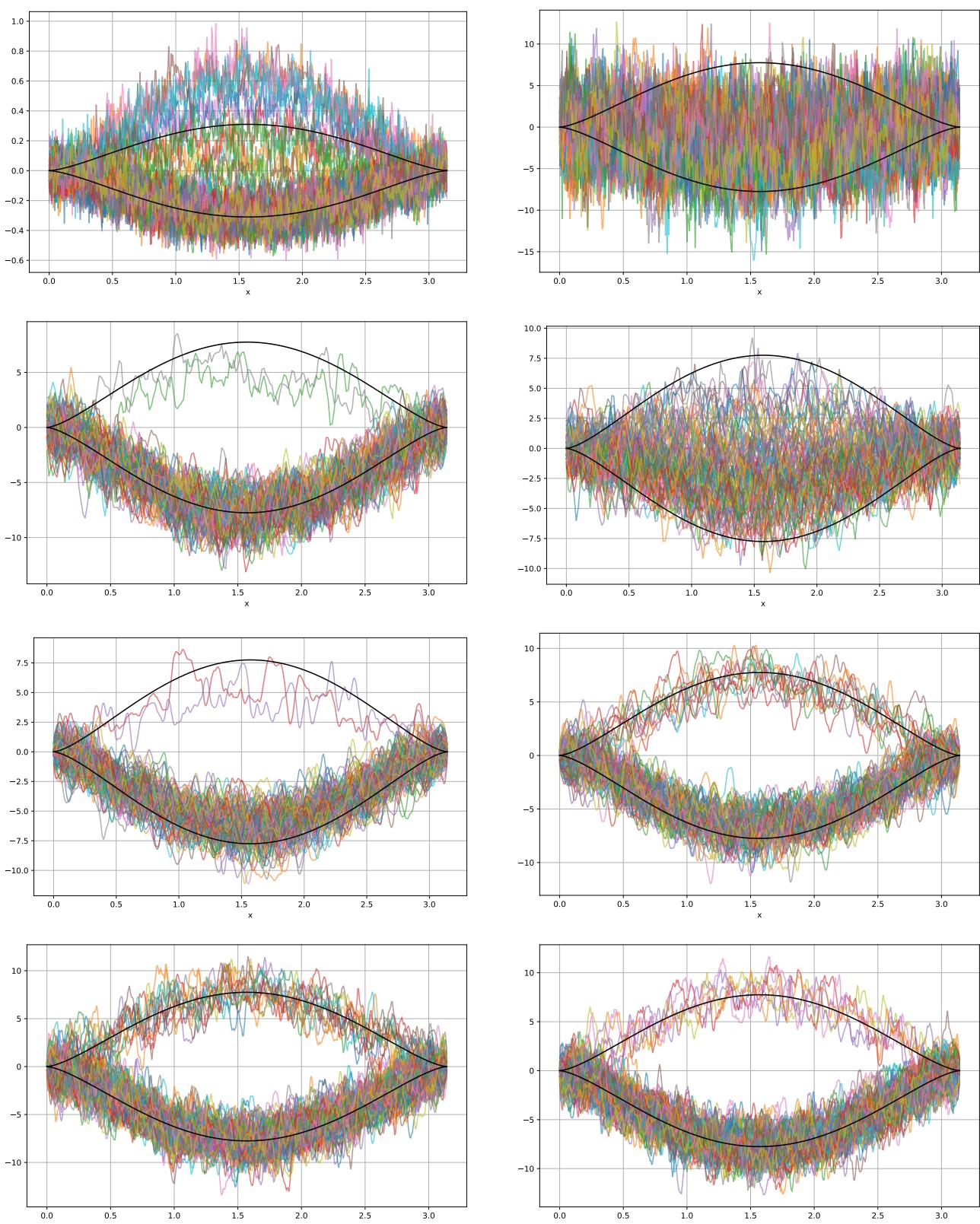

*Figure 5.* **Gaussian mixture** target as defined in (11) sampled through various generative diffusion models. **First row:** the finite-dimensional denoising SDE. **Second row:** the infinite-dimensional denoising SDE with constant noise. **Third row:** our forced VP-SPDE with constant noise. **Fourth row:** our forced VP-SPDE with noise scheduling. **Column one:** noising horizon $T = 1$. **Column two:** noising horizon $T = 0.2$.

---

**Algorithm 2** Sampling Forced Generative VP-SPDE

---

**Require:** Approximated steering function $s_\theta$, covariance operator $C$, numerical SDE solver solve
**Require:** Langevin steps $N_{\text{Langevin}}$, step size $\epsilon_{\text{Langevin}}$, integration steps $N_{\text{solve}}$
  1: **Prior Sampling of** $\mu_0^h$
  2: Initialize $X_0^h \sim \mathcal{N}(0, C)$
  3: **for** $k = 1$ **to** $N_{\text{Langevin}}$ **do**
  4:      Sample noise $\eta \sim \mathcal{N}(0, C)$
  5:      Apply pCN-Langevin update:

$$X_0^h \leftarrow \sqrt{1 - \epsilon_{\text{Langevin}}^2} X_0^h + \frac{\epsilon_{\text{Langevin}}^2}{2} C s_\theta(0, X_0^h) + \epsilon_{\text{Langevin}} \eta$$

  6: **end for**
  7: **Time Integration of** $X^h$
  8: Integrate the forced diffusion model over $N_{\text{solve}}$ steps:

$$X^h \leftarrow \text{solve}(X_0^h, A, s_\theta, Q, N_{\text{solve}})$$

  9: **return** $X^h$

---

## F.2. Gaussian mixture example 7.1

**Model and training** All methods are implemented using the same neural network (NN) architecture to approximate the score/steering function. As NN we use a fully connected FiLM-conditioned residual score network that maps an input $x \in \mathbb{R}^D$ to the target function in $\mathbb{R}^D$ at hand. The time argument is injected into each residual block, processed after a sinusoidal time embedding through a multilayer perceptron. The architecture consists of an input projection to hidden width $H$ and $B$ LayerNorm-MLP residual blocks. The hidden width $H$ and number of blocks $B$ are chosen as functions of the input dimensions $D$. In the largest case that we consider, $D = 500$, the total NN size consists of roughly $10e8$ parameters.

The learned reverse time and forced diffusions are integrated using a semi-implicit Euler-Maruyama (IEM) scheme with 250 steps. In the cases that we use a Langevin sampler to target the initial distribution of the forced diffusions, we take 50 Langevin steps.

For the noise-varying models we use a linear noise schedule $\beta(t) = \beta_0 + t(\beta_1 - \beta_0)$ with $\beta_0 = 0.1$ and $\beta_1 = 20$. For the forced diffusions, a time reversed version $\tilde{\beta}(t) = \beta(1 - t)$ is used. The constant noise schedules are fixed at a constant $\beta = 10$ to match the total amount of noise injected over $[0, 1]$.

## F.3. MNIST-SDF example 7.2

**Data** We use here the MNIST-SDF dataset, in which training samples are generated by converting MNIST digit images into continuous signed distance functions (Sitzmann et al., 2020). A representation in 'SDF space' is displayed in Figure 6(a). This representation allows for consistent training and evaluation across resolutions, while generated samples can be thresholded back into binary digits by applying a mask for comparison with standard MNIST-based metrics, see Figure 1.

**Model and training** For the sake of comparability, we adopt the NN architecture proposed in (Lim et al., 2025) to approximate the steering term $s(t, x)$. This consists of a Fourier Neural Operator (FNO) based U-net with three up- and downsampling stages, a base width of $64$ channels and channel multipliers $(1, 2, 2)$, using $4$ spectral res blocks per stage. All spatial convolutions are replaced by spectral convolutions, with group normalisation performed in Fourier space based on a fixed number of modes. Downsampling and upsampling are implemented via alias-free filtered resampling (Karras et al., 2021) and the architecture excludes self-attention and dropout. More architectural details can be found in (Lim et al., 2025), Appendix J.

The forced VP-SPDE (7) is implemented with a time-reversed cosine noise schedule. The covariance matrix $C$ is constructed by taking the empirical marginal variances of the dataset in Fourier space for the first 32 modes and scaling the remaining modes $j > 32$ appropriately by a decay rate of $j^{-(1+\varepsilon)}$ such that $C$ remains trace-class. The model is trained for a total of $2 \times 10^6$ iterations on upsampled $64 \times 64$ MNIST-SDF images. At inference time, the trained model is numerically integrated

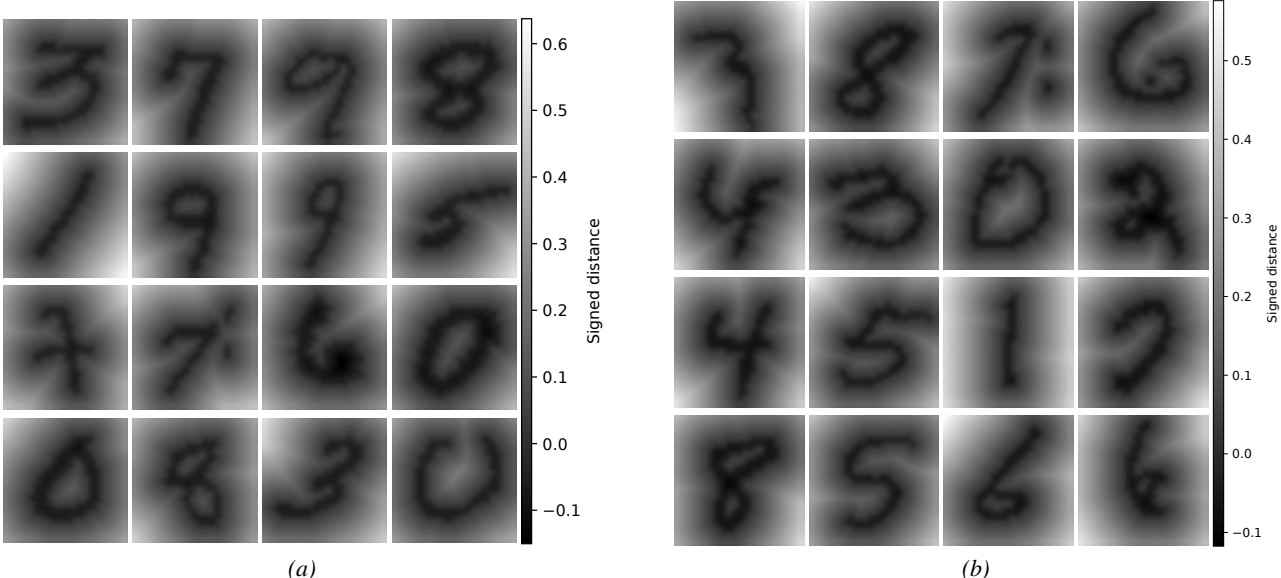

*Figure 6.* True and generated **MNIST-SDF** samples at $64 \times 64$ resolution. **(a):** True samples, upsampled to $64 \times 64$. **(b):** Generated samples returned by the VP-SPDE model.

based on a semi-implicit Euler-Maruyama scheme with 250 steps. Following (Lim et al., 2025), FID scores are computed after applying a binary mask to the true and generated samples.

**Resolution Independence**  The forced VP-SPDE model (8) as well as the loss function (6) are formulated entirely in the function space setting and are therefore inherently resolution independent. In practice, after approximating the steering term $s(t, x)$ by a resolution independent network $s_\theta(t, x)$ such as the FNO U-net proposed in (Lim et al., 2025), Appendix J, images at higher resolutions - unseen during training - may be generated.

Figure 8 shows MNIST-SDF images generated by the forced VP-SPDE model (8) at a resolution of $128 \times 128$, based on the score trained on $64 \times 64$ resolution images. Moreover, Table 4 provides the FID score obtained by our model in comparison to those reported in (Lim et al., 2025). In our experiments we have encountered high frequency Gibbs ringing of the learned steering term $s_\theta(t, x)$ applied at this resolution, facilitating the need to wrap the score network in another down- and upsampling step during numerical integration of the model. We hypothesize that these resolution-induced artifacts stem from the specific neural network architecture rather than our underlying theoretical framework, and we postpone investigations into artifact-free architectures within our framework to future research.

### F.4. Bayesian inverse problem: seismic imaging 7.3

**Data**  We use the data provided by (Baldassari et al., 2023) from their open-source repository[1], consisting of synthetic training pairs based on seismic images from the Kirchhoff-migrated Parihaka-3D dataset (Veritas, 2005; WesternGeco, 2012).

**Models**  The score approximation of the forced diffusion model is based on a Fourier Neural Operator architecture that maps an input field $x$, a conditioning observation $y$ and spatial coordinates to a single-channel output. The continuous time variable $t$ is embedded using Gaussian Fourier features followed by an MLP layer, with the embedding injected into each

---

[1]https://github.com/alisiahkoohi/csgm

*Table 4.* FID scores on MNIST-SDF at $128 \times 128$ resolution. The values marked with $^\star$ have been reported in (Lim et al., 2025).

|  | GANO | MultiDiff | DDO | VP-SPDE (ours) |
|---|---|---|---|---|
| FID | $13.05^\star$ | $201.08^\star$ | $\mathbf{7.96}^\star$ | 15.66 |

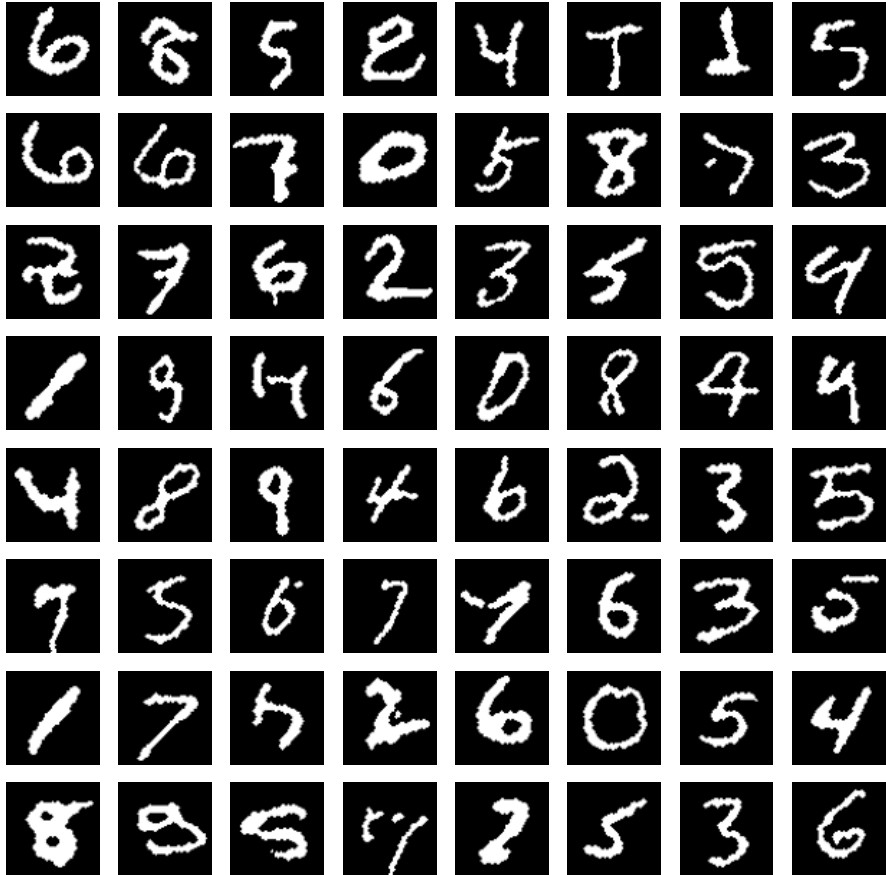

*Figure 7.* Additional generated **MNIST-SDF** samples at $64 \times 64$ resolution.

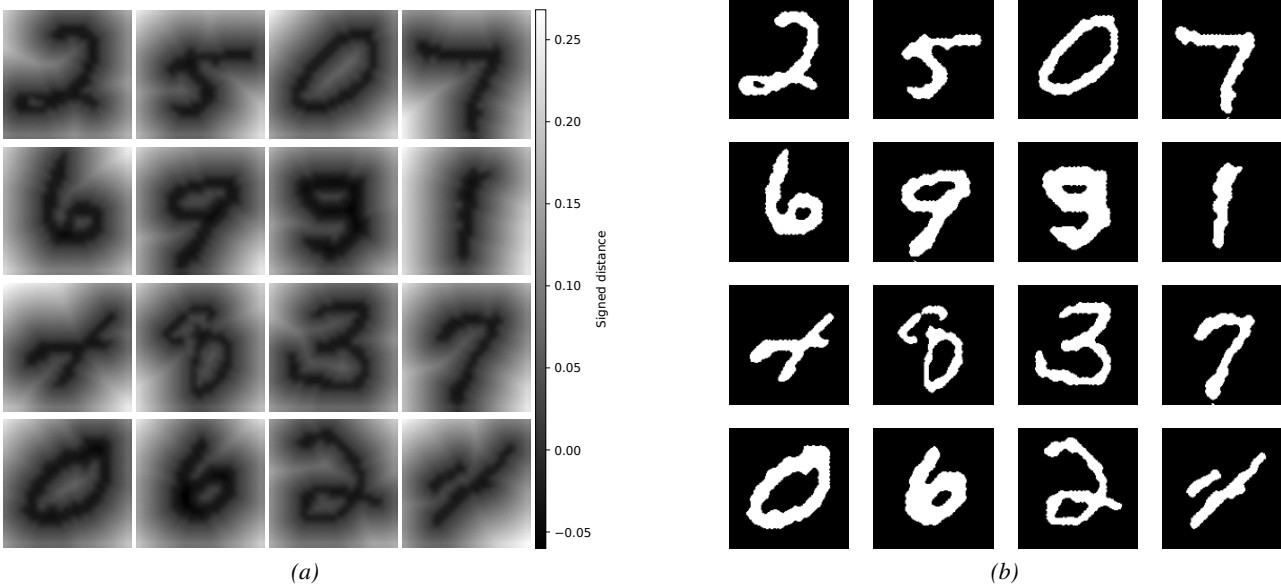

*Figure 8.* Generated **MNIST-SDF** samples at $128 \times 128$ resolution. **(a):** Samples in function space. **(b):** Masked samples.

FNO layer via FiLM modulation. The network comprises a pointwise lifting layer, four Fourier neural layers operating on the 24 lowest Fourier modes with pointwise skip connections, and a final pointwise projection to the output.

The network is trained for $50\,000$ iterations with a training batch-size of 128, which amounts to 676 training epochs on the given dataset. For the VP-SPDE, we use a linear noise schedule $\beta(t) = \beta_1 + t(\beta_0 - \beta_1)$ with $\beta_0 = 0.1$ and $\beta_1 = 20$ and, following (Baldassari et al., 2023), a white noise covariance operator $C$. During sample generation, the forced VP-SPDE (8) is numerically solved using a semi-implicit Euler-Maruyama scheme with 500 steps.

For the implementation of the noising-denoising model, we follow the parametrisation and procedure as outlined in (Baldassari et al., 2023), Appendix D. In particular, we keep a total of 500 discrete noise levels, aligning with the number of EM steps in our continuous time model. For the sake of comparability, the number of training epochs is increased from 300 in the original work to 676, matching the number of training iterations used for our model.

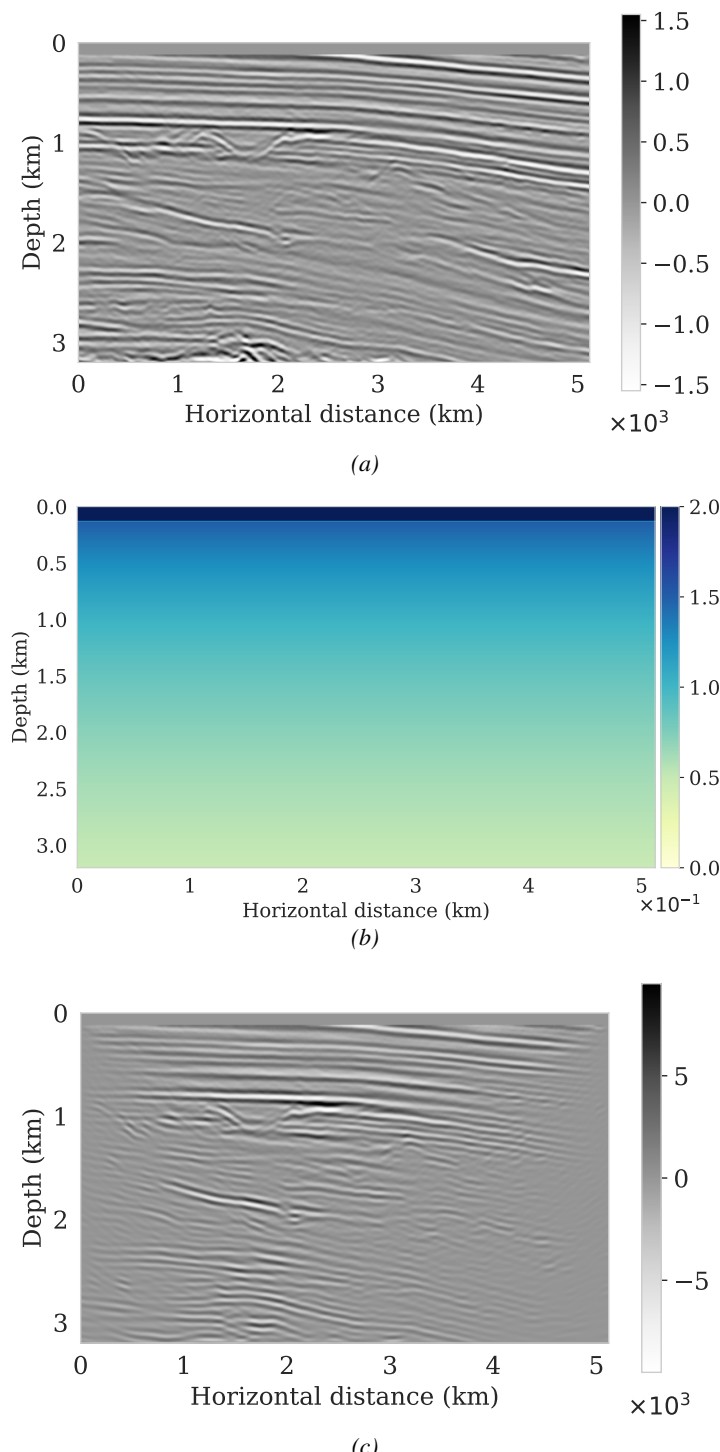

*Figure 9.* **Seismic imaging.** **(a)** Ground truth seismic image $x$. **(b)** Background smooth squared slowness model. **(c)** Observed data $y$ after application of the linearised, adjoint Born operator.

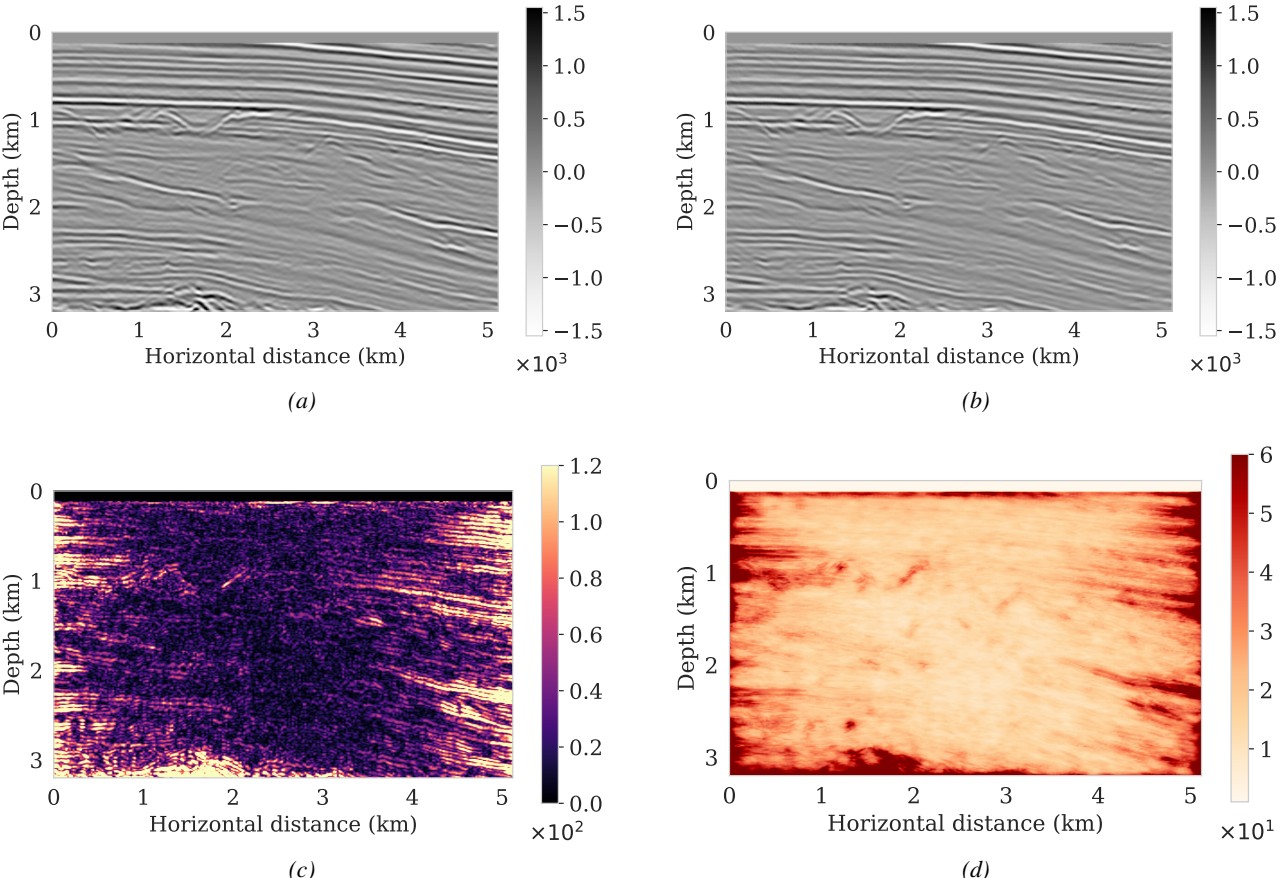

*Figure 10.* **Seismic imaging estimate** generated by the model of (Baldassari et al., 2023). **(a)** Ground truth seismic image $x$. **(b)** Estimated posterior mean $\hat{x}$. **(c)** Absolute error between ground truth and posterior mean. **(d)** Estimated marginal posterior standard deviation.

