# OpenReview forum: "Infinite-Dimensional Generative Diffusions via Doob’s h-Transform"
_ICML.cc/2026/Conference — ICML 2026 regular_

### Official Review · Reviewer_q2aj · 2026-02-24

**Soundness:** 3
**Presentation:** 3
**Significance:** 3
**Originality:** 3
**Overall Recommendation:** 4
**Confidence:** 3

**Summary:**

This paper introduces a principled framework for defining generative diffusion models in infinite dimensions based on Doob’s h-transform, without requiring time reversal. Moreover, the method is well-defined for arbitrary time horizons and is evaluated on both synthetic and real-world infinite-dimensional data.

**Compliance With Llm Reviewing Policy:**

Affirmed.

**Final Justification:**

The authors have thoroughly addressed all my concerns, so I recommend accepting this paper.

**Key Questions For Authors:**

The paper would benefit from clearer exposition and a more complete background. To improve accessibility, I recommend including essential background information to help readers better understand the contributions.

Overall, I consider this to be a solid and valuable paper, and I will adjust my score based on the feedback from other reviewers.

**Limitations:**

Yes.

**Strengths And Weaknesses:**

*Strengths*

+ This paper pioneers the application of the Doob h-transform to infinite-dimensional generative modeling, circumventing the path-theoretic shortcomings inherent in traditional reverse SDEs. It provides a unified framework for handling diverse regularity scenarios, featuring rigorous theoretical derivations and established error bounds. This constitutes a highly solid work.

+ The mathematical derivations in this paper are exceptionally rigorous. The appendix provides complete proofs for all claims made in the main text and thoroughly discusses the assumptions.

+ The visualization analysis in Figure 2 fully demonstrates the effectiveness of the proposed method under a mixed Gaussian distribution, providing experimental evidence for the approach presented in this paper.

*Weaknesses*

+ The experimental evaluation is not sufficiently comprehensive, as it is limited to two small-scale datasets: Gaussian mixture and MNIST-SDF. In contrast, the compared methods [1] were evaluated on a broader range of tasks, including the Volcano dataset, etc. To strengthen the claims and demonstrate the generalizability of the proposed method, the authors should include experiments on additional, larger-scale datasets.

   >  [1] Score-based Diffusion Models in Function Space.

+ Additionally, in Section 6, the authors transition from theoretical analysis to method design but omit detailed algorithm descriptions. I recommend adding this content, as it is essential for bridging the gap between theory and practical implementation.

+ In line 44, it would be best to summarize the shortcomings arising from the finite-dimensional discretization of inherently infinite-dimensional objects, which is the problem addressed in this paper. Moreover, this paper primarily explores problems in infinite-dimensional spaces; however, the introduction does not clearly specify the core challenges in comparison to finite-dimensional settings. Including illustrative examples would make the paper more accessible.

---

> ### Author Rebuttal · Authors · 2026-03-31
>
> Dear Reviewer q2aj,
>
> we are very grateful for your positive feedback on our work and for your suggestions regarding a more complete background exposition.
> Below, we address the weaknesses raised in your review and detail the specific revisions we have made to strengthen the manuscript.
>
> **On Experiments.**
> We agree that a weakness of the submitted manuscript is the lack of more comprehensive experiments.
> We have therefore decided to include another experiment on an infinite-dimensional Bayesian inverse problem motivated by a seismic imaging problem introduced in [1]. Here, the aim is to infer the high-frequency components of the Earth's subsurface squared-slowness model, conditioned on surface seismic observations.
> Our methodology naturally adapts to this setting by learning a conditional h-transform, training the steering function on joint samples (X,Y) of the target distribution and observations. At inference time, given an observation $Y = y$, the learned model guides the forced diffusion process toward the posterior distribution $X \mid Y = y$.
>
> As detailed in the table below, our approach manages to outperform the current state-of-the-art noising-denoising baseline in the literature ([1]) by achieving lower relative and mean absolute reconstruction errors on data unseen during training while yielding a better-calibrated representation of posterior uncertainties. The accurate recovery of these complex geological structures  demonstrates that our method performs effectively on larger scale datasets and highly complex inverse problems.
>
> | Method | Relative error $\|\hat{x}-x\|/\|x\|$ | Avg. abs. error | Avg. poster. std. |
> |:---|:---:|---:|---:|
> | VP-SPDE (ours) | 0.09 | 26.82 | 35.28 |
> | Noising–denoising | 0.17 | 37.80 | 24.27 |
>
> **On Algorithm Description.**
> In the final version, we include algorithmic summaries of our methodologies training and sampling procedures. These will help readers better understand the practical contributions of our work.
>
> **On Infinite-Dimensional Challenges.**
> As part of the introduction, we detail the challenges of infinite-dimensional generative models in Section 1.1. These include the lack of a Lebesgue measure, the choice of noising process $W$ and the difficulties of obtaining well-defined time reversals.
>
>
>
> [1] Lorenzo Baldassari, Ali Siahkoohi, Josselin Garnier, Knut Solna, and
> Maarten V de Hoop. Conditional score-based diffusion models for bayesian
> inference in infinite dimensions. Advances in Neural Information Processing
> Systems, 36:24262–24290, 2023.

---

> > ### Author Rebuttal · Reviewer_q2aj · 2026-04-01
> >
> > I thank the authors for their detailed response, which has thoroughly addressed my concerns. Given the paper's solid theoretical analysis and empirical results, I have decided to maintain my positive score.

---

### Official Review · Reviewer_SLT6 · 2026-03-05

**Soundness:** 3
**Presentation:** 3
**Significance:** 2
**Originality:** 3
**Overall Recommendation:** 4
**Confidence:** 4

**Summary:**

This paper presents a framework for generative modeling in infinite-dimensional spaces using Doob’s h-transform. Rather than following the standard diffusion paradigm of separate forward noising and reverse denoising stages, the authors formulate the generative process as a direct forward simulation of a forced diffusion bridge. This bridge is constructed by twisting a reference dynamics with a drift term derived from an h-function, which steers the process toward the data distribution. The authors provide a theoretical treatment of the system’s well-posedness, derive a corresponding training objective, and establish formal error bounds for the approximation. The method is empirically validated through a toy Gaussian mixture model with an analytical solution and a higher-dimensional MNIST-SDF generation task.

**Compliance With Llm Reviewing Policy:**

Affirmed.

**Final Justification:**

Given the clarity provided and the technical merit of the work, I am now confident in recommending this paper for acceptance and look forward to seeing the updated results in the final version.

**Key Questions For Authors:**

1. In Proposition 4.1, the loss is defined as an expectation under the target bridge measure $P^h$. Since the true h-function is unknown during training, how do you practically sample from this measure to compute the loss?
2. In the Gaussian mixture experiments, are the learned bridges sampled using functional decompositions or coordinate-wise evaluations on a fixed grid? If it is the latter, how does this framework fundamentally differ from a high-dimensional finite setting in practice?
3. Regarding the claim in Line 359 that "finite-dimensional methods diverge as D increases," could you clarify if the baseline models were scaled appropriately? Specifically, were the neural network capacities (number of parameters) and discretization steps kept consistent across all values of D? Without this parity, the observed divergence may stem from insufficient model capacity or numerical instability in the baseline rather than an inherent limitation of finite-dimensional frameworks.
4. For the MNIST task, to fully evaluate the necessity of an infinite-dimensional setting, it is important to see "resolution-free" capabilities. Can the model generate samples at arbitrary coordinates or resolutions that were not explicitly seen during training (e.g., 128×128)?

**Limitations:**

Yes

**Strengths And Weaknesses:**

**Strengths**

The manuscript is generally well-written, accessible, and provides a clear motivation for addressing the challenges of generative modeling in infinite-dimensional spaces. A key strength of the work is the rigorous connection it establishes between infinite-dimensional SPDEs and traditional denoising models. The authors provide a formal derivation for the explicit form of the forced SPDE, specifically for the Variance Preserving (VP) case, which adds significant theoretical value. Furthermore, the inclusion of a detailed error bound analysis for the proposed score-matching objective strengthens the mathematical foundation of the framework and provides important guarantees for the method's convergence and stability.

**Weaknesses**

Despite the technical execution, the paper suffers from a significant oversight in its literature review that impacts the assessment of its originality. The core mathematical framework for forcing infinite-dimensional diffusion processes via Doob's h-transform has been previously established in existing literature[1], where bridge processes were defined via an exponential change of measure and optimized using a similar score-matching objective. The lack of discussion regarding these highly relevant works makes it difficult to ascertain the unique contribution of this submission.

Beyond the literature concerns, the experimental validation is somewhat limited and does not fully substantiate the paper's primary "infinite-dimensional" claims. The Gaussian mixture model uses high but ultimately finite dimensions (D=500 only), and the MNIST generation (only on 64x64)—does not sufficiently demonstrate the power of an infinite-dimensional formulation over standard high-dimensional discrete models. To truly showcase the advantages of the proposed approach, evaluations on truly functional data or more complex infinite-dimensional tasks would be necessary. Finally, the experimental details provided in the appendix are insufficient, which poses a challenge for the reproducibility of the results and limits the ability of other researchers to verify the authors' empirical findings.

[1] Baker, E., et.al., Conditioning non-linear and infinite-dimensional diffusion processes, NeurIPS 2024

---

> ### Author Rebuttal · Authors · 2026-03-31
>
> Dear Reviewer SLT6,
>
> we thank you for your careful evaluation of our manuscript and for highlighting limitations in our experiments as well as an important reference.
>
> Based on your questions regarding the literature and our method, it appears our framing may have caused some confusion regarding the scope of our method, in particular with respect to the difference between sampling diffusion bridges and generative modeling via diffusion bridges.
> Below, we address the weaknesses raised and answer your key questions.
>
> **1. Regarding connection to diffusion bridges**
>
> We would first like to address the claim of a `significant oversight in the literature'. It is true that conditioning infinite-dimensional diffusions via Doob's h-transform has previously been treated in the literature. Results of this nature have already been established in [1], two decades before [2,3] - all three we cite in our introduction.
>
> Specifically, [2] addresses the problem of ___learning an infinite-dimensional diffusion bridge___, in which a fixed - typically non-linear - diffusion process is conditioned to hit a predefined point (one `observation') $y \in H$ at time T >0 (or, in [2], a set of positive measure - a more tractable case, as one does not condition on a null set).
>
> If the dynamics are linear, the diffusion bridge can be derived analytically as the Ornstein-Uhlenbeck bridge. If the dynamics are non-linear, the problem becomes difficult since the bridge process - derived by Doob's h-transform - includes a steering term based on the intractable transition densities of the non-linear diffusion. In [2], a score-matching approach is applied to approximate the steering term by learning two time-reversals.
>
> In contrast, our work concerns ___generative modeling___, in which one aims to `learn' a sampler that targets an unknown distribution $\mu$, given iid samples $y_1, \ldots y_N$ thereof. Except in the trivial case that $\mu$ is a Dirac measure, this is a fundamentally different problem.
> We address it by conditioning a reference diffusion (of which we have free choice, in contrast to bridge problems) by Doob's h-transform to satisfy $\mu$ at time T. Likewise to the problem above, this also includes an intractable steering term. However, the steering term is intractable due to the unknown $\mu$, not due to the non-linear dynamics of the reference diffusions.
>
> **2. Regarding experimental validation**
>
> We agree that a weakness of the submitted manuscript is the lack of more comprehensive experiments, and therefore include, in our final version, an additional experiment on an infinite-dimensional Bayesian inverse problem. Crucially, this experiment highlights that our framework scales effectively while outperforming the current state-of-the-art in both relative error and variance consistency. Please refer to our response to Reviewer q2aj for more details.
>
> In terms of the finite-dimensionality of the Gaussian example, we ended up choosing $D = 500$ since this was sufficient to showcase that a `naive', finite-dimensional implementation of a diffusion model diverges (cf. Figure 4).
>
> **Answers to Key Questions for Authors.**
>
> **Question 1**
>
> We explain how to sample from $P^h$ during training in Remark 3.4 and Remark 4.2, based on the given training data $y_1, \ldots, y_N$.
>
> **Question 2**
>
> The methods are implemented based on a spectral decomposition defined by the eigenbasis of the Mat\'ern covariance kernel $C$, rather than a coordinate-wise evaluation on a grid. We now added a remark to the relevant section to clarify this.
>
> **Question 3**
>
> We are unsure if we correctly understand your question here. The network architecture is dependent on the dimension D. However, between the various methods, the same network is implemented. We'd like to stress that the comparison between the methods and the divergence of a finite-dimensional method is not a question of network- but model choice. A `naive' white noise diffusion will lead to an ill-defined loss function since marginals with respect to Lebesgue measure are non-existent in the limit $D \to \infty$.
>
> **Question 4**
>
> The forced generative diffusion framework is formulated entirely in the function space setting and the underlying Fourier neural network architecture is compatible with generating images of arbitrary resolution. Given the time constraints, we are not able to demonstrate this now, but we will include a demonstration in the final version.
>
> [1]Marco Fuhrman. A class of stochastic optimal control problems in hilbert
> spaces: Bsdes and optimal control laws, state constraints, conditioned pro-
> cesses. 108(2):263–298, 2003.
>
> [2] Elizabeth L Baker, Gefan Yang, Michael L Severinsen, Christy A Hipsley, and
> Stefan Sommer. Conditioning non-linear and infinite-dimensional diffusion pro-
> cesses. 37:10801–10826, 2024.
>
> [3]Thorben Pieper-Sethmacher, Frank van der Meulen, and Aad van der Vaart.
> On a class of exponential changes of measure for stochastic pdes. 185:104630, 2025

---

> > ### Author Rebuttal · Reviewer_SLT6 · 2026-04-03
> >
> > I would like to thank the authors for their thorough and patient rebuttal. After carefully reviewing the clarifications, I realize that my initial negative recommendation was based on a misunderstanding of the specific generative problem and bridge sampling setup. This led me to undervalue the paper's core contribution. The authors' replies have addressed my questions and concerns.
> >
> > While the current experimental scope is somewhat limited—a point the authors themselves acknowledge—I now recognize the theoretical and methodological value of the work, and I believe this would be beneficial to the community. I have therefore raised my recommendation score accordingly.

---

### Official Review · Reviewer_gHAz · 2026-03-09

**Soundness:** 4
**Presentation:** 3
**Significance:** 3
**Originality:** 3
**Overall Recommendation:** 5
**Confidence:** 4

**Summary:**

This paper introduces a new method for generative modelling in (infinite-dimensional) Hilbert spaces. Instead of using a noising SDE and learning the time-reversal, the paper suggests learning a bridge from a reference distribution (e.g. a Gaussian) to the data distribution directly. This is formulated in Hilbert spaces under the assumption that the data distribution is absolutely continuous with respect to a Gaussian (although, the paper also discusses the extension to data distributions within the Cameron-Martin space of a Gaussian measure instead). The paper also considers an infinite-dimensional analogue of the variance preserving SDE. Errors between the true measure and the learned measure are provided.

**Compliance With Llm Reviewing Policy:**

Affirmed.

**Final Justification:**

I think the paper has good theoretical contributions. The rebuttal also somewhat addressed my concern about the experiments by conducting another experiment and benchmarking against an existing method, where they get good results. Therefore I continue to recommend the paper's acceptance.

**Key Questions For Authors:**

1. Under which circumstances does one expect the data measure not to be noised to a Gaussian "quickly" (as with the point of Weaknesses 3.) ?

2. In Section 3 what’s $\mu_0$? For example are there any assumptions on it such as being absolutely continuous with respect to $\nu$, or that samples can be obtained in closed form?

3. In Section 2 is it correct that Assumption 2.1 is equivalent to situation (A) in Section 1.1 or is there some difference? If these are equivalent it could be useful to the reader to point this out.

4. If one uses samples from $\mu_0$ instead of $\mu_0^h$ in the SDE (4) can this be seen as somehow equivalent to the method of e.g. Pidstrigach et al. 2024, where it is assumed that at time $T$ the distribution is approximately Gaussian?

5. Related to Weaknesses 4., how expensive is it to sample from $\mu^h$ e.g. how many steps are typically needed?

**Limitations:**

Yes.

**Strengths And Weaknesses:**

**Strengths:**
1. One of the main strengths of the paper is in the writing and narrative. I find it very well structured and easy to understand, despite the subject matter being quite technical - I think the paper is well-written.
2. In a similar vein, the paper is technically sound with all assumptions clearly stated. Moreover, the assumptions made are reasonable and appropriately discussed. All claims are theoretically supported.
3. The work addresses a gap in the literature, providing a new method for diffusion models on function spaces that does not rely on the approximation of X_T being approximately distributed by a Gaussian for some large T, as is the case with time-reversals, since the method instead forces the SDE to have a Gaussian (or other chosen) distribution at the end (or technically at the start) time.

**Weaknesses:**

I don’t think there are any critical weaknesses in the paper, however, I do think there is some room for improvement.

1. One potential weakness is the positioning of the paper within the literature. Although I think many of the main necessary papers are cited I think it would make sense to compare to [1] (and perhaps some of the succeeding related works like [2]). This is formulated in finite dimensions where the maths is quite different, so does not impact the originality of this paper. However, I believe [1] can be viewed as a finite dimensional version of the presented method and therefore should be discussed. Moreover, I think it would make sense to cite [3] which again has a different focus but seems related. For this reason, I have scored the paper a 3 on presentation instead of a 4.
2. Experiments: The experiment section is somewhat limited with experiments including multivariate Gaussians and MNIST-SDF, however I don’t find this to be a major issue since I think the paper provides nice theoretical contributions.
3. Significance: One of the improvements of this method over prior works is when the standard Ornstein-Uhlenbeck noising SDE does not converge to noise very quickly. However, I think the manuscript would be improved with more clarifications on when this is expected to happen in practice. An experiment on such a situation would further strengthen this point.
4. Clarity on implementation: I would appreciate more discussion on sampling from $\mu_0^h$. I understand that the paper uses an infinite-dimensional Langevin sampler, which is cited and is not the paper's contribution, however I still would find more details on this useful, since the paper claims it is a fundamental feature of the proposed methodology.
I also wonder whether this could be an expensive step and in that sense wonder how it compares to choosing a larger $T$ in the time-reversal method.


The following are minor and do not influence my recommendation:
1. Line 098: “abstrahizes” is not a word. Perhaps “abstracts” is meant?
2. Line 176: “due insufficient noising time T” should maybe be “due to insufficient noising at time T”.
3. Line 359 column 2: “and those generated the …”
4. Line 240: “exist” should be “exists”.
5. Line 322 column 2: “withconstant”
6. In general, within the references the capitalisation and formatting should be checked and fixed. A few examples are: “schr\”odinger”, “Mcmc”, “Bsdes”

**Summary**

In summary, I recommend this paper be accepted. I find it to be well-written with good theoretical contributions. The paper could be further improved with stronger experiments and benchmarking which I believe could improve the significance, by further demonstrating in which situations this method is advantageous to others. However, I find the theoretical content of the paper is strong enough without this.


[1] Peluchetti https://arxiv.org/pdf/2312.14589

[2] Liu et al. https://arxiv.org/pdf/2208.14699

[3] Baker et al. https://proceedings.neurips.cc/paper_files/paper/2024/file/14ad9256c430e6c8977e470d8e268320-Paper-Conference.pdf

---

> ### Author Rebuttal · Authors · 2026-03-31
>
> Dear Reviewer gHaz,
>
> we thank you for the positive feedback, your thorough assessment of our manuscript and for the detailed technical questions.
>
> With respect to positioning of the manuscript in the literature, we would like to refer you to our reply to Reviewer Dxk1, where we have included remarks on a related work section that we will include in the final version. These include references that you have pointed out to us.
>
> Moreover, we agree that a weakness of the submitted manuscript is the lack of more comprehensive experiments, and therefore include, in our final version, an additional experiment on an infinite-dimensional Bayesian inverse problem. Crucially, this experiment highlights that our framework scales effectively while outperforming the current state-of-the-art in both relative error and variance consistency. Please refer to our response to Reviewer q2aj for more details.
>
> Below we provide detailed answers to your key questions:
>
> **Question 1**
>
> The standard approach in the noising-denoising framework is to ensure that sufficient noise has been injected in the system by a large enough T > 0.
> The Gaussian nature of the noising process makes it easy to compute relevant statistics such as the signal-to-noise ratio to verify this.
> However, choosing T too large increases the computational burden and introduces sampling bias due to increased discretisation errors overs the extended integration time, hence motivating research on how to optimally choose T such as in [1].
> In contrast, the approach via forcing completely circumvents this issue.
>
> **Question 2**
>
> The results in Section 3 are presented in a general form and $\mu_0$ can, on paper, be any Borel measure such that Assumption 3.1 holds true for the diffusion X.
> However, for the sake of the application outlined in Section 4, it is practical to set $\mu_0$ as either a Dirac- or a Gaussian measure, absolutely continuous with respect to $\nu$, to keep the loss function easy to compute. This is described in Remark 4.2. From Section 5 onwards we simply set $\mu_0 = \nu = \mathcal{N}(0,C)$. This is similar to the noising-denoising spirit in which the noising process converges to $\mathcal{N}(0,C)$.
> An interesting research direction involves the case where $\mu_0$ is a fixed Dirac measure and noise is injected purely through the Wiener process at time of sample generation. However, we hypothesize that this reduces the regularity of the steering function as $t \to 0$, and hence will lead to a model that is harder to train; we leave the empirical investigation of this phenomenon to future work.
>
> **Question 3**
>
> This is indeed the case and we have added a remark to point this out to the reader. Likewise, we have done so for the connection between Section 6.1 and `situation (B)' in Section 1.1.
>
> **Question 4**
>
> Even for a sufficient noising time T > 0, where it can be assumed that $\mu_0^h \approx \mu_0$ and that the noising process in [2] has converged, the two methods remain distinct in nature. For example, even in the simple case of a Gaussian target measure, the tractable closed form expressions of the steering terms will differ.
>
> **Question 5**
>
> One Langevin sampling step is effectively as expensive as one time integration step of the forced diffusion model. The number of Langevin steps should therefore be carefully weighed against the number of time integration steps at sample generation time.
> A principled way to determine the required Langevin steps is to initialise multiple parallel chains at $\mu_0$ and compute the Gelman-Rubin diagnostics $\hat{R}$ as a function of the number of steps (as test function, we can use the potential of $\mu_0$, for example). Typically  $\hat{R}<1.05$ indicates that convergence to the stationary distribution has occurred.
> In our examples, due to sufficient noising time, we have found the necessity of including Langevin steps only in the Gaussian mixture experiment when setting $T = 0.2$.
>
>
>
> [1] Giulio Franzese, Simone Rossi, Lixuan Yang, Alessandro Finamore, Dario
> Rossi, Maurizio Filippone, and Pietro Michiardi. How much is enough? a
> study on diffusion times in score-based generative models. Entropy, 25(4):633,
> 2023
>
> [2] Jakiw Pidstrigach, Youssef Marzouk, Sebastian Reich, and Sven Wang.
> Infinite-dimensional diffusion models. Journal of Machine Learning Research,
> 25(414):1–52, 2024.

---

> > ### Author Rebuttal · Reviewer_gHAz · 2026-03-31
> >
> > I thank the authors for their response. I did not have any major concerns about the paper and continue to recommend for the paper's acceptance.

---

### Official Review · Reviewer_Dxk1 · 2026-03-13

**Soundness:** 3
**Presentation:** 3
**Significance:** 3
**Originality:** 2
**Overall Recommendation:** 4
**Confidence:** 2

**Summary:**

This paper develops a framework for diffusion models on infinite dimensional Hilbert spaces. Motivated by the fact many data types are inherently infinite-dimensional, a recent line of research has focused on adapting diffusion models to that specific setting. The. underlying goal is to avoid discretization as much as possible in the modelling process to develop methods that are robust and generalize well across different resolutions.

On finite dimensional space, the core theoretical framework relies on the approximation of a reverse time stochastic differential equation which contained the so-called score function in its drift. This score function writes as the density with respect to the Lebesgue measure of the time marginal of some hand-designed forward (noising) process. However, in infinite dimension such construction does not extend directly, since there is no canonical counterpart of the Lebesgue measure.

Contrary to works that try to construct a time-reversed SDE, as in the finite dimensional case, the authors suggest not to rely on time reversal but rather to leverage Doob's $h$-transform. This enables to construct a process that is guaranteed to reach the target measure at an arbitrary final time $T$. The resulting SDE closely mirror standard score-based generative models (SGMs), with the distinction the score function is replaced by a "steering term" defined as the gradient of the log of the $h$-transform. The authors also propose a method to approximate this term using a score matching objective that mimics the finite-dimensional SGM training procedure, and can be interpreted as a mixture-of-bridge representation.

Finally the paper introduces a theoretical decomposition for the 2-Wasserstein distance and illustrates the approach empirically on function-space examples.

**Compliance With Llm Reviewing Policy:**

Affirmed.

**Key Questions For Authors:**

**1. Relation to stochastic optimal control formulations.**
The paper develops the infinite-dimensional generative diffusion framework using Doob’s h-transform. However, diffusion bridges and Schrödinger-type problems are often closely related to stochastic optimal control formulations. Could the authors clarify the relationship between their construction and recent stochastic control formulations of diffusion models (for example adaptive denoising diffusions or stochastic control formulations of diffusion bridges in function spaces)?
- Can the proposed construction be interpreted as a stochastic control problem?
- Does the h-transform correspond to an optimal control value function in some cases?
- Are there advantages of the Doob-transform formulation compared to the control formulation?

**2. Relation to recent work on infinite-dimensional diffusion bridges.**
There has been recent work on infinite-dimensional diffusion bridges and conditioning of stochastic processes (e.g., operator-learning approaches and conditioning frameworks). Could the authors clarify more explicitly how their approach differs from these works and what new insight is obtained from the Doob-transform perspective?

**3. Scope of applicability of the framework.**
Could the authors clarify under what conditions the proposed framework applies (e.g., classes of SPDEs, Gaussian reference processes, or particular function spaces)? It would also be useful to understand what the main practical limitations of the approach are.

**Limitations:**

yes

**Strengths And Weaknesses:**

**Soundness.**
The paper appears technically sound. The proposed framework based on Doob’s h-transform for constructing infinite-dimensional generative diffusion processes is mathematically well grounded and fits naturally within the theory of conditioned stochastic processes and diffusion bridges. The assumptions appear reasonable for the infinite-dimensional setting and the arguments seem consistent with the established literature on stochastic analysis and diffusion conditioning.

One aspect that could further strengthen the paper would be a clearer discussion of the relationship between this Doob-transform construction and the stochastic optimal control perspective on diffusion models, which has recently emerged as an important theoretical viewpoint. In particular, diffusion bridges and Schrödinger-type formulations are often closely connected to stochastic control formulations, and clarifying how the present framework relates to this literature would improve the conceptual positioning of the work.

Relevant related directions that could be discussed include recent works connecting diffusion models, diffusion bridges, and stochastic control in infinite-dimensional or functional settings (e.g., Christensen et al., 2024; Park et al., 2024; Yang et al., 2024; Baker et al., 2024). Making these connections explicit would help clarify whether the Doob h-transform provides an alternative formulation, a generalization, or a complementary viewpoint.

**Presentation.**
The paper is generally well written and clearly structured. The mathematical development is logically organized and the progression from background material to the main construction is easy to follow.

That said, I found it somewhat difficult to precisely assess the degree of novelty of the contribution due to the limited discussion comparing the approach to closely related works on diffusion bridges, conditioning of infinite-dimensional diffusions, and operator-learning approaches to such problems. A more explicit comparison with these related approaches would improve clarity regarding what is fundamentally new versus what is a reformulation of existing constructions.

This is mainly a positioning issue rather than a clarity issue.

**Significance.**
The paper addresses a relevant theoretical problem. It contributes to the ongoing effort to better understand the mathematical structure of diffusion-based generative models beyond finite-dimensional settings and to the use of Doob's h-transform in this literature. The potential impact is therefore mostly conceptual and theoretical, but still meaningful for the development of principled generative modeling in function spaces.

**Originality.**
The originality appears reasonable and aligned with recent developments connecting diffusion models with classical stochastic process theory. In particular, the use of Doob’s h-transform as a generative modeling tool fits well within a broader trend of reinterpreting diffusion models through conditioning, bridges, and Schrödinger-type formulations.
The contribution can therefore be viewed as part of a broader emerging line of work connecting diffusion models with conditioning techniques and stochastic analysis.

_References suggested for positioning_
- Sören Christensen, Jan Kallsen, Claudia Strauch, Lukas Trottner. Beyond Fixed Horizons: A Theoretical Framework for Adaptive Denoising Diffusions.
- Byoungwoo Park, Jungwon Choi, Sungbin Lim, Juho Lee. Stochastic Optimal Control for Diffusion Bridges in Function Spaces. (2024).
- Gefan Yang, Elizabeth L. Baker, Michael L. Severinsen, Christy A. Hipsley, Stefan Sommer. Infinite-dimensional diffusion bridge simulation via operator learning. (2024).
- Elizabeth L. Baker, Gefan Yang, Michael L. Severinsen, Christy A. Hipsley, Stefan Sommer. Conditioning non-linear and infinite-dimensional diffusion processes. (2024).

---

> ### Author Rebuttal · Authors · 2026-03-31
>
> Dear Reviwer Dxk1,
>
> thank you for the supportive comments and for directing our attention to related literature. They have significantly helped us contextualize our contributions. To answer your questions:
>
> **Questions 1 & 2**
>
> Our final version includes a related work section that positions our work within the literature on DSB, SOC and sampling of diffusion bridges. For the connection to diffusion bridges, please refer to our response to Reviewer SLT6.
>
> To better position our framework, we explicitly related our approach to the literature on finite-dimensional schrödinger diffusion bridges and optimal control. Specifically, we contextualize our work with respect to Diffusion Schrödinger Bridges (DSB) and SOC methods [8, 3, 4, 6], forward-time and non-denoising frameworks like Diffusion Bridge Mixture Transport (DBMT) [1, 2, 10], and recent formulations utilizing Doob's h-transforms [5].
>
> To the best of our knowledge, the only comparable result that extends the rich literature on generative diffusion bridges to the infinite-dimensional setting is given in [9]. Here, Ornstein-Uhlenbeck bridges - a linear diffusion pinned on an initial and terminal state - are derived from a SOC perspective and subsequently used to extend the DBMT approach of [1]. The methodology is restricted to time autonomous, diagonalisable diffusions. Moreover, no existence results for the diffusion bridge mixture or bounds to the target are given.
>
> In contrast, our work provides an alternative probabilistic framework that defines a generative diffusion, forced to sample exactly from a target distribution $\mu$, based on one unified change of measure $P^h$. This change of measure perspective is particularly useful in the infinite-dimensional setting, as it can be understood as a Girsanov-type transformation under which existence of mild solutions to SPDEs are well understood in the existing literature [7].
>
> In practice, this offers a wider flexibility in the choice of the reference diffusion X compared to [1, 9], allowing the forcing of non-autonomous or even non-linear diffusions and SPDEs with unbounded drift operators.
> As an additional consequence, existence results of the forced diffusion as a mild solution to an infinite-dimensional equation can be stated under verifiable assumptions on the target $\mu$ and the reference process X. This allows us to obtain 2-Wasserstein bounds between the target measure and the learned measure.
>
> Our current framework prioritizes the construction of a generative diffusion model based on a learnable change of measure. The literature on finite-dimensional models indeed suggests a rich connection between our approach and the SOC and DSB perspectives. This represents a promising future research direction, in particular in solving an infinite-dimensional DSB without time-reversals as well as connecting our h-function of choice to the solution of a stochastic optimal control problem.
> However, as a rigorous mathematical treatment of this connection is highly demanding, it falls outside the scope of the present manuscript.
>
> **Question 3**
> Conditions for the target $\mu$ and reference X require only Assumptions 2.1 and 3.1, with no extra restrictions on SPDE classes or function spaces. The rather abstract Assumption 3.1 is verifiable via Proposition 5.1 and Corollary 5.5.
> However, in practice, using a linear diffusion for X is highly advantageous; tractable transition densities and closed-form Ornstein-Uhlenbeck bridges drastically reduce the loss function's computational cost. While linear noising remains standard, our framework provides the flexibility to support future non-linear diffusions in infinite dimensions.
>
>
> [1] Stefano Peluchetti. Diffusion bridge mixture transports, schr¨odinger bridge
> problems and generative modeling.
>
> [2] Stefano Peluchetti. Non-denoising forward-time diffusions.
>
> [3] Yuyang Shi, Valentin De Bortoli, Andrew Campbell, and Arnaud Doucet. Dif-
> fusion schrödinger bridge matching.
>
> [4] Alexander Tong, Nikolay Malkin, Kilian Fatras, Lazar Atanackovic, Yan-
> lei Zhang, Guillaume Huguet, Guy Wolf, and Yoshua Bengio. Simulation-
> free schrödinger bridges via score and flow matching
>
> [5] Sören Christensen, Jan Kallsen, Claudia Strauch, and Lukas Trottner. Beyond
> fixed horizons: A theoretical framework for adaptive denoising diffusions.
>
> [6] Tianrong Chen, Guan-Horng Liu, and Evangelos A Theodorou. Likelihood
> training of schrödinger bridge using forward-backward sdes theory.
>
> [7] Giuseppe Da Prato and Jerzy Zabczyk. Stochastic equations in infinite dimen-
> sion.
>
> [8] Valentin De Bortoli, James Thornton, Jeremy Heng, and Arnaud Doucet. Dif-
> fusion schrödinger bridge with applications to score-based generative modeling.
>
> [9] Byoungwoo Park, Jungwon Choi, Sungbin Lim, and Juho Lee. Stochastic op-
> timal control for diffusion bridges in function spaces.
>
> [10] Xingchao Liu, Lemeng Wu, Mao Ye, and Qiang Liu. Let us build bridges:
> Understanding and extending diffusion generative models.

---

> > ### Author Rebuttal · Reviewer_Dxk1 · 2026-04-03
> >
> > Thank you for the detailed responses and clarifications provided during the rebuttal phase. The explanations regarding the theoretical aspects and implementation details satisfactorily addressed my questions and improved the clarity of the contribution.
> >
> > Based on these clarifications, I confirm my positive assessment and maintain my original score.

---

### Decision · Program_Chairs · 2026-04-30

**Decision:**

Accept (regular)

**Comment:**

All reviewers have voted in favor of acceptance for this paper, noting its sound theoretical contributions. While there were some concerns regarding the scope of the experiments, and that some of the practical aspects were not fully developed, the authors address these concerns during the rebuttal period. I highly encourage the authors to incorporate the detailed feedback received from the reviewers in the discussion process in an updated version of their work.

#### Hallucinated References
Please note that the following reference was flagged as potentially being hallucinated:

Reference: Trippe, B. L., Yim, J., Torge, D., Watson, J., Barzilay, R., and Jaakkola, T. A framework for conditional diffusion modelling with applications in protein design and inverse problems. arXiv preprint arXiv:2312.09236, 2024.

After checking this reference myself it does not seem to exist. Please be sure fix this for the next version of the paper.